# $D^3$: Dynamic Directional Graph-Constrained Data Scheduling for LLM Training

**Yuanjian Xu** [1 2 †]  **Jianing Hao** [1]  **Guang Zhang** [1 *]  **Zhong Li** [2 *]

## Abstract

Training data plays a central role in large language models (LLMs) optimization, motivating extensive research on data scheduling strategies. Most existing approaches concentrate on adjusting the overall data distribution but neglect the underlying interactions between samples during training. However, we argue that such interactions cannot be overlooked, as real-world data samples frequently exhibit directional influences on each other, making the training order crucial. Intuitively, we can prioritize train-units with greater influence to improves learning efficiency. In this work, we propose $D^3$, a **D**ynamic, **D**irectional graph-constrained **D**ata scheduling framework. $D^3$ formulates the complex interactions among train-units as a dynamic influence graph, where edges represent loss-based dependencies. It then solves a constrained optimization problem over this graph to derive the training order, which ensures that the data sequence respects the evolving information flow throughout training. Our approach is theoretically motivated and yields consistent improvements over existing data scheduling methods across both pre-training and post-training phases. Furthermore, for scalability, $D^3$ also employs an efficient approximation algorithm that keeps the additional computational overhead within a manageable range. For future research, the code is available at https://github.com/xuyj233/D3.

## 1. Introduction

High-quality training data remains the primary driver behind the empirical success of large language models

(LLMs) (Hsieh et al., 2023; Muennighoff et al., 2023; Bharathi Mohan et al., 2024; Villalobos et al., 2024). Recent work has shown that intrinsic properties of training data, including data quality, distributional composition, and sample difficulty, shape the learning trajectory and downstream generalization performance of LLMs (Xu et al., 2023; Hoffmann et al., 2022; Xie et al., 2023a). Consequently, a plethora of data-centric strategies, including selection, filtering, and reweighting, have been proposed to curate more effective training distributions (Sorscher et al., 2022; Xie et al., 2023b; Albalak et al., 2024). Despite their disparate formulations, these methods share a common objective: to enhance model robustness and generalization by optimizing over a set of admissible data distributions $\mathcal{Q}$, rather than the raw empirical distribution (Sagawa et al., 2019).

However, a key mechanism governing data utility, which involves the dynamic interactions of the train-unit (defined as inter-batch or inter-sample interactions, where the latter is a degenerate case of the former with a batch size of 1) interactions that evolve throughout training, has been largely overlooked. The efficacy of a train-unit depends not only on its intrinsic properties but also on when it is presented relative to others, as early train-units can prime the model for later ones, creating dependencies that render the data non-exchangeable. Recent empirical studies confirm that data ordering substantially influences convergence and final performance in LLM training (Kim & Lee, 2024; Pouransari et al., 2024; Jia et al., 2025). Yet these scheduling approaches remain largely heuristic; they lack a principled, optimization-grounded method to quantify the very interactions they seek to exploit. This leads to a core open question: how can we formally model dynamic train-unit interactions to derive optimal data schedules?

To address this gap, we introduce $D^3$, a framework that reformulates data scheduling as a constrained optimization problem over a dynamic graph. We start from the principle that the optimal order should prioritize train-units that most effectively "prepare" the model for subsequent data. $D^3$ makes this precise by modeling pairwise influence via look-ahead loss reduction: if training sample $i$ now significantly lowers the future loss of train-unit $j$, then $i$ should precede $j$. By estimating these directional effects throughout training, $D^3$ maintains a dynamic influence graph that captures the evolving topology of train-unit dependencies. The training

*Co-corresponding authors. †Work done during Yuanjian Xu's internship at Microsoft Research Asia. [1]HKUST-GZ [2]Microsoft Research. Correspondence to: Guang Zhang <guangzhang@hkust-gz.edu.cn>, Zhong Li <zhongli@microsoft.com>.

*Proceedings of the 43$^{rd}$ International Conference on Machine Learning*, Seoul, South Korea. PMLR 306, 2026. Copyright 2026 by the author(s).

sequence is then derived as the solution that best respects this graph's structure, ensuring the model follows a coherent, interaction-aware learning trajectory. Our contributions are summarized as follows:

- We propose $D^3$, a data scheduling framework that formalizes sequence ordering as an optimization problem over a dynamic training influence graph.

- By providing a fundamental method to characterize asymmetric train-unit interactions, we reveal that these relationships dictate the necessity of a structured training sequence, addressing a significant gap in conventional distribution-centric approaches.

- Extensive numerical validations across multiple pre-training and post-training tasks show that $D^3$ consistently improves the performance of LLMs with reasonable computation overheads.

## 2. Related Work

We organize related work into three categories: (i) sample-level data selection (reweighting) methods, (ii) domain-level reweighting approaches, and (iii) empirical studies that investigate the effects of data ordering during LLM training.

### 2.1. Domain-Level Data Mixture Optimization

Domain-level optimization determines the optimal composition of macroscopic data sources (e.g., web, code, academic text) for pretraining, moving beyond simple heuristics. Inspired by scaling laws (Kaplan et al., 2020), recent studies introduce automated, optimization-driven frameworks. A seminal approach is DoReMi (Xie et al., 2023a), which uses a small proxy model to derive domain weights via distributionally robust optimization. Subsequent methods enhance efficiency and scalability: REGMIX (Liu et al., 2025) treats the problem as a regression task using micro-models, MixMin (Thudi et al., 2025) formalizes mixture finding as a convex minimization problem. Notably, MixMin demonstrates that optimal mixtures found on small models can be scale-invariant. To reduce retraining costs, Chameleon (Xie et al., 2025) utilizes leverage scores in a learned embedding space for flexible domain reweighting, and Velocitune (Luo et al., 2025) adaptively adjusts weights based on learning velocity, specifically targeting the complexities of domain-adaptive continual pre-training. In comprehensive empirical studies, Shukor et al. (2025) propose scaling laws for optimal data mixtures, further establishing the theoretical foundation for these data-centric strategies.

**Sample-level Data Strategies.** Sample-level strategies prioritize individual training instances, offering finer-grained control than domain-level methods. A key recent advance is the shift toward dynamic, online selection mechanisms. GREATS (Wang et al., 2024) applies greedy optimization via Taylor expansion to select high-quality batches in every iteration, while Sow et al. (2025a) introduces online loss-based reweighting to focus training on more informative samples. To capture the complex interactions between data points, Group-MATES (Yu et al., 2025b) proposes group-level selection using a relational data influence model. Another paradigm leverages training artifacts and optimization frameworks to refine data importance. AutoMixer (Chang et al., 2025) utilizes checkpoint models to approximate data influence, whereas DWM (Yu et al., 2025a) implements a bi-level optimization framework to update a Data Weighting Model dynamically. Furthermore, to resolve potential conflicts between heterogeneous selection criteria, Multi-Actor Collaboration frameworks (Bai et al., 2025) have been proposed to integrate multiple prioritization rules through a central console. However, most existing sample-level methods either incur heavy computational overhead due to second-order optimizations or lack a unified geometric principle to guide the data selection process, particularly during critical phases such as model annealing.

### 2.2. Data Curriculum in LLM training

Previous methods have empirically validated the importance of sample ordering in LLM training (Kim & Lee, 2024; Pouransari et al., 2024; Jia et al., 2025). Orca (Mukherjee et al., 2023) establishes a progressive learning paradigm that leverages complex explanation traces to enhance reasoning capabilities. Recent research on contrastive post-training shows that a curriculum transitioning from easy preference pairs to hard ones yields step-function improvements in model alignment (Zhou et al., 2024). Despite these empirical successes, most curriculum strategies remain grounded in heuristic designs. There is still a lack of fundamental understanding regarding the underlying training dynamics, particularly how asymmetric gradient interactions between samples influence the optimization trajectory.

## 3. Methodology

In this section, we begin with a formal formulation of learning with data scheduling. In Section 3.2, we introduce the construction of an influence graph, which captures how loss curvatures induce asymmetric dependencies among training units. Subsequently, in Section 3.3, we formulate the data scheduling task as a constrained optimization problem governed by the influence graph. Section 3.4 presents an efficient solver for this problem. Finally, the implementation of $D^3$ is detailed in Section 3.5, which focuses on the practical approximations designed to ensure scalability and efficiency in large-scale training settings.

## 3.1. Problem Formulation

We formalize one epoch of LLM training as a discrete dynamical system over the parameter space $\Theta \subseteq \mathbb{R}^d$. Given a training dataset $\mathcal{D} = \{z_i\}_{i=1}^N$, we consider a fixed partition into $K$ disjoint batches $\mathcal{P} = \{\mathcal{B}_1, \ldots, \mathcal{B}_K\}$ of equal size $B = N/K$ (assuming $N$ is divisible by $K$). The empirical risk on a batch $\mathcal{B} \in \mathcal{P}$ is defined as: $\ell(\mathcal{B}; \theta) = \frac{1}{B} \sum_{z \in \mathcal{B}} \ell(z; \theta)$. A data schedule is specified by a permutation $\pi \in \mathcal{S}_K$, where $\mathcal{S}_K$ denotes the symmetric group of degree $K$. When $B = 1$ (hence $K = N$), the schedule operates at the sample level; for $B > 1$, it characterizes batch-level scheduling. At each discrete step $t \in \{1, \ldots, T\}$, where $T = K$ for one epoch, the model state is updated by a stochastic gradient operator $\mathcal{T}_{\pi(t)} : \Theta \to \Theta$:

$$\theta^{(t)} = \mathcal{T}_{\pi(t)}\big(\theta^{(t-1)}\big) = \theta^{(t-1)} - \eta \, \nabla_\theta \, \ell\big(\mathcal{B}_{\pi(t)}; \theta^{(t-1)}\big),$$

where $\eta > 0$ is the learning rate. The terminal state after processing all $K$ batches is the composition:

$$\theta^{(T)}(\pi) = \big(\mathcal{T}_{\pi(T)} \circ \mathcal{T}_{\pi(T-1)} \circ \cdots \circ \mathcal{T}_{\pi(1)}\big)\big(\theta^{(0)}\big).$$

Because gradient operators are generally non-commutative in non-convex landscapes, the terminal state $\theta^{(T)}(\pi)$ is path-dependent: different permutations $\pi$ lead to different final parameters and hence different generalization performance. The data scheduling problem aims to find a permutation that minimizes the expected risk on a test distribution $\mathcal{D}_{\text{test}}$:

$$\pi^* = \arg \min_{\pi \in \mathcal{S}_K} \mathbb{E}_{z \sim \mathcal{D}_{\text{test}}} \Big[ \ell\big(z; \theta^{(T)}(\pi)\big) \Big].$$

We emphasize that this is a combinatorial optimization over $K!$ possible schedules, which is intractable for large $K$. Our work seeks efficient algorithms to approximate $\pi^*$ by exploiting the interaction between training batches.

## 3.2. The Construction of Influence Graph

To characterize the interactions among training batches during optimization, we introduce the concept of training influence prediction. This measure, formally defined below, predicts how the loss of one training unit would change if the model were updated using the gradient of another one.

**Definition 3.1** (Training Influence Prediction). Let $\theta^{(t)}$ be the model parameters at optimization step $t$. Consider a hypothetical update using the gradient from batch $\mathcal{B}_i$:

$$\theta_i^{(t+1)} := \theta^{(t)} - \gamma \, \nabla_\theta \, \ell(\mathcal{B}_i; \theta^{(t)}),$$

where $\gamma > 0$ is a look-ahead parameter that controls the step size for influence estimation. The training influence of batch $\mathcal{B}_i$ on batch $\mathcal{B}_j$ at step $t$ is defined as:

$$\mathcal{I}_{i \to j}(t) := \ell\big(\mathcal{B}_j; \theta_i^{(t+1)}\big) - \ell\big(\mathcal{B}_j; \theta^{(t)}\big).$$

By varying $\gamma$, this formulation allows us to probe interactions at different temporal scales: small values of $\gamma$ reflect short-term and local effects, while larger values of $\gamma$ approximate accumulated and long-term influences.

Intuitively, more negative influence values correspond to greater loss decrement on $\mathcal{B}_j$; resulted from the gradient descent on $\mathcal{B}_i$. We have the following second-order approximation of the training influence in Proposition 3.2, which shows that training influence is symmetric to the first order but becomes asymmetric when curvature effects are considered.

**Proposition 3.2** (Local Structure of Influence Prediction). *Assume that $\ell(\cdot; \theta)$ is twice differentiable. A second-order Taylor expansion gives*

$$\mathcal{I}_{i \to j}(t) = -\gamma \underbrace{\nabla_\theta \ell(\mathcal{B}_j; \theta^{(t)})^\top \nabla_\theta \ell(\mathcal{B}_i; \theta^{(t)})}_{\mathcal{I}_{i \to j}^{(1)}(t)}$$

$$+ \frac{\gamma^2}{2} \underbrace{\nabla_\theta \ell(\mathcal{B}_i; \theta^{(t)})^\top \nabla_\theta^2 \ell(\mathcal{B}_j; \theta^{(t)}) \nabla_\theta \ell(\mathcal{B}_i; \theta^{(t)})}_{\mathcal{I}_{i \to j}^{(2)}(t)}$$

$$+ o(\gamma^2).$$

Obviously, the first-order influence term is pairwise symmetric: $\mathcal{I}_{i \to j}^{(1)}(t) = \mathcal{I}_{j \to i}^{(1)}(t)$, while the second-order influence term is asymmetric: $\mathcal{I}_{i \to j}^{(2)}(t) \neq \mathcal{I}_{j \to i}^{(2)}(t)$, due to the per-batch Hessian $\nabla_\theta^2 \ell(\mathcal{B}_j; \theta^{(t)})$.

At the same time, since $\mathcal{I}_{i \to j}(t)$ depends on the current parameter state $\theta^{(t)}$, the above interaction evolves throughout optimization, naturally inducing a dynamic and directed relation among training samples, which can be further characterized as a dynamic directed graph structure.

**Definition 3.3** (Influence Graph). At step $t$, the influence graph is a directed weighted graph $\mathcal{G}^{(t)} = (\mathcal{V}, \mathcal{E}^{(t)})$, where each node $i \in \mathcal{V}$ corresponds to a training batch $\mathcal{B}_i$. A directed edge $(i, j) \in \mathcal{E}^{(t)}$ exists if $\mathcal{I}_{i \to j}(t) \neq 0$, with weight $\mathcal{I}_{i \to j}(t)$. The graph evolves over time through its dependence on $\theta^{(t)}$. Equivalently, $\mathcal{G}^{(t)}$ can be encoded by an adjacency matrix $\mathbf{A}^{(t)} \in \mathbb{R}^{K \times K}$ with entries $\mathbf{A}_{ij}^{(t)} = \mathcal{I}_{i \to j}(t)$.

## 3.3. Influence Graph-Constrained Optimization

The influence graph $\mathcal{G}^{(t)}$ (Definition 3.3) captures asymmetric dependencies among batches, but it does not directly prescribe an ordering for sequential training. To determine an optimal sequence, we transform the graph into a linear chain that respects strong directional preferences while resolving conflicts from cycles. We formalize these preferences using a directional advantage matrix that quantifies the relative benefit of training one batch before another.

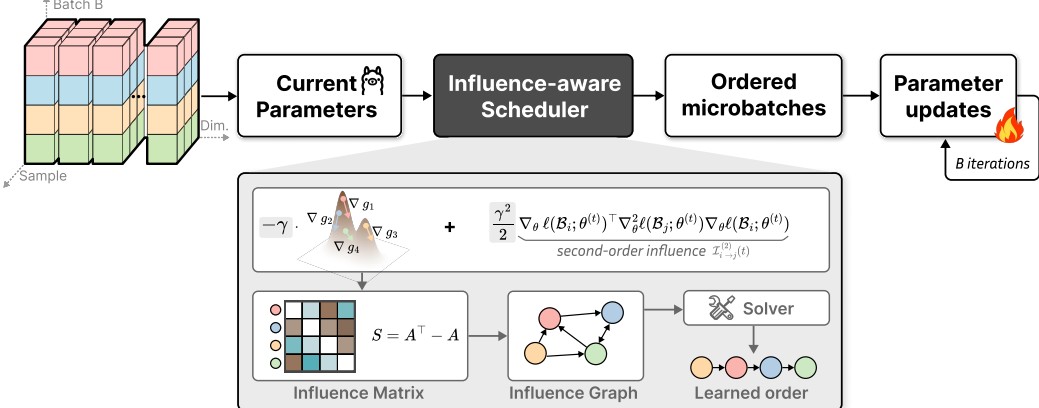

*Figure 1.* **Overview of the $D^3$ scheduling framework.** Given $B$ micro-batches and the current parameter state $\theta^{(t)}$, the scheduler evaluates pairwise interactions by computing batch gradients and second-order curvature terms. These interactions are formalized into a directional influence matrix $\mathbf{S}^{(t)}$, which induces a dynamic influence graph. A graph-based solver then optimizes the sequence over this graph to produce the learned training order $\pi^{(t)}$, which is applied during SGD updates to facilitate a more efficient optimization trajectory.

**Definition 3.4** (Influence Advantage Graph). Given the influence graph $\mathcal{G}^{(t)}$ with adjacency matrix $\mathbf{A}^{(t)}$, we define the influence advantage graph as the same structure augmented with a directional advantage matrix

$$\mathbf{S}^{(t)} := \mathbf{A}^{(t)\top} - \mathbf{A}^{(t)},$$

where $\mathbf{S}_{ij}^{(t)} = \mathcal{I}_{j\to i}(t) - \mathcal{I}_{i\to j}(t)$ quantifies the advantage of training $\mathcal{B}_i$ before $\mathcal{B}_j$.

An edge $(i, j)$ with positive directional advantage $\mathbf{S}_{ij}^{(t)} > 0$ corresponds to $\mathcal{I}_{i\to j}(t) < \mathcal{I}_{j\to i}(t)$, indicating that training batch $\mathcal{B}_i$ before $\mathcal{B}_j$ yields greater cumulative loss improvement than the reverse order. The set of dominance constraints is therefore

$$\mathcal{E}_{\text{dom}}^{(t)} := \{(i, j) \mid \mathbf{S}_{ij}^{(t)} > 0\} = \{(i, j) \mid \mathcal{I}_{i\to j}(t) < \mathcal{I}_{j\to i}(t)\}.$$

Transforming the fully connected influence advantage graph into a linear chain incurs a cost. Each edge with positive directional advantage that is contradicted by the ordering contributes a penalty.

Formally, let $\pi^{(t)} \in \mathcal{S}_K$ be a batch ordering, where $\pi^{(t)}(s) = k$ indicates that batch $\mathcal{B}_k$ is placed at position $s$. Define its inverse mapping $\sigma^{(t)} : \{1, \ldots, K\} \to \{1, \ldots, K\}$ by $\sigma^{(t)}(k) = s$ whenever $\pi^{(t)}(s) = k$; then $\sigma^{(t)}(k)$ gives the position of batch $\mathcal{B}_k$. Both $\pi^{(t)}$ and $\sigma^{(t)}$ are bijections and belong to $\mathcal{S}_K$, representing the same ordering. Using $\sigma^{(t)}$ as the decision variable, the scheduling problem minimizes the total penalty from violated directional advantages:

$$\sigma^{(t)\star} = \arg\min_{\sigma \in \mathcal{S}_K} \sum_{i \neq j} \max\!\left(0, \mathbf{S}_{ij}^{(t)}\right) \cdot \mathbb{I}[\sigma(j) < \sigma(i)].$$

If $\mathbf{S}_{ij}^{(t)} > 0$ (i.e., edge $(i, j)$ favors $i$ before $j$) but $\sigma(j) < \sigma(i)$ in the ordering, the edge is violated and incurs cost $\mathbf{S}_{ij}^{(t)}$. When the graph is acyclic, there exists an ordering with zero cost, corresponding to a topological sort of the graph; otherwise, cycles force trade-offs, and the optimal ordering minimizes the total violation penalty.

### 3.4. Solving the Scheduling Optimization

We propose a lightweight solver, Random-Swap Refinemen (RSR), to approximately minimize the violation cost. The solver operates in two stages: first, it computes an initial ordering based on the advantage matrix row sums; second, it refines this ordering through random pairwise swaps.

**Initial ordering via row sums.** For each batch $i$, compute the row sum $r_i = \sum_{j=1}^K \mathbf{S}_{ij}^{(t)}$. Since a positive entry $\mathbf{S}_{ij}^{(t)}$ indicates that placing $i$ before $j$ is beneficial, a larger $r_i$ suggests that batch $i$ has a global advantage when scheduled earlier. We therefore sort the batches in decreasing order of $r_i$ to obtain an initial permutation $\pi_0$.

**Random-swap refinement.** Starting from $\pi_0$, we perform up to $M$ random pairwise swap trials. In each trial, we randomly choose two distinct positions $p$ and $q$ (with $p < q$) and consider swapping the batches at these positions. Let $i = \pi(p)$ and $j = \pi(q)$. The change in violation cost $\Delta\mathcal{C}$ due to swapping $i$ and $j$ can be computed efficiently by observing that only the relative order of $i$ and $j$ with respect to each other and to the batches between them changes. Specifically, let $I(p, q) = \{\pi(k) \mid p < k < q\}$ be the set of batches between positions $p$ and $q$. Then:

$$\Delta\mathcal{C} = \mathbf{S}_{ij}^{(t)} + \sum_{k \in I(p,q)} \left(\mathbf{S}_{ik}^{(t)} + \mathbf{S}_{kj}^{(t)}\right).$$

If $\Delta\mathcal{C} < 0$, the swap is accepted and $\pi$ is updated accordingly.

**Complexity analysis.** While this random-swap heuristic does not guarantee global optimality, it provides a computationally efficient method to construct a high-quality ordering that respects most directional advantages. The initial ordering requires computing all row sums and sorting, which takes $O(K^2)$ time. Each swap trial evaluates $\Delta\mathcal{C}$ in $O(|I(p,q)|)$ time, bounded by $O(K)$ in the worst case. Hence, the total time for $M$ trials is $O(MK)$, and the overall complexity of the solver is $O(K^2 + MK)$.

### 3.5. The Detailed Implementation of $D^3$

Efficient training influence scheduling for LLMs must address two scaling issues: a large sample count $N$ and high-dimensional gradients $\mathbb{R}^d$ (with $d \sim 10^9$–$10^{12}$). Computing pairwise influences and solving the exact ordering problem is infeasible at this scale. We therefore introduce two approximations: chunk-wise internal reordering and gradient compression for influence propagation.

**Chunk-wise Internal Reordering.** To scale to large $K$, we adopt a stochastic chunking strategy. At each step $t$, we randomly sample $L \ll K$ batches to form a chunk $\mathcal{C}^{(t)}$. The two-stage solver is then applied within $\mathcal{C}^{(t)}$ to reorder these $L$ batches based on their $L \times L$ advantage sub-matrix. Batches outside the chunk keep their relative order. This reduces per-step complexity from $O(K^2)$ to $O(L^2 + ML)$ and ensures all batches are gradually refined over time through repeated sampling.

**Gradient information compression.** To handle extreme gradient dimensionality, we project each batch gradient into a low-dimensional manifold using a fixed Gaussian random matrix $\mathbf{R} \in \mathbb{R}^{d \times k}$ ($k \ll d$). This Johnson–Lindenstrauss projection approximately preserves inner products with high probability (Achlioptas, 2001). For the gradient of batch $\mathcal{B}_i$, denoted as $\mathbf{g}_i^{(t)} = \nabla_\theta \ell(\mathcal{B}_i; \theta^{(t)})$, we compute its low-dimensional sketch $\tilde{\mathbf{g}}_i^{(t)} = \mathbf{R}^\top \mathbf{g}_i^{(t)} \in \mathbb{R}^k$. The first-order influence term can then be efficiently approximated as:

$$\mathcal{I}_{i \to j}^{(1)}(t) = \left(\mathbf{g}_j^{(t)}\right)^\top \mathbf{g}_i^{(t)} \approx \left(\tilde{\mathbf{g}}_j^{(t)}\right)^\top \tilde{\mathbf{g}}_i^{(t)}.$$

This compressed inner product allows us to compute the advantage matrix entries $\mathbf{S}_{ij}^{(t)}$ with $O(dk + k)$ memory and time per pair, instead of $O(d)$.

**Hessian approximation.** For the second-order term, we avoid the full Hessian by adopting a diagonal approximation via Hutchinson estimation (Liu et al., 2024). Using $W$ Hutchinson vectors $\mathbf{u}^{(w)} \sim \mathcal{N}(0, \mathbf{I}_d)$, we estimate the diagonal Hessian for batch $\mathcal{B}_j$ as

$$\hat{\mathbf{h}}_j^{(t)} = \frac{1}{W} \sum_{w=1}^{W} \mathbf{u}^{(w)} \odot \left(\nabla_\theta^2 \ell(\mathcal{B}_j; \theta^{(t)}) \mathbf{u}^{(w)}\right),$$

where $\odot$ denotes the Hadamard product. To avoid storing the $d$-dimensional vector $\hat{\mathbf{h}}_j^{(t)}$ for every batch, we project the curvature onto the gradient direction, obtaining a scalar measure:

$$\lambda_j^{(t)} = \left(\mathbf{g}_j^{(t)}\right)^\top \mathrm{diag}\left(\hat{\mathbf{h}}_j^{(t)}\right) \mathbf{g}_j^{(t)}.$$

We then approximate the second-order influence term as $\mathcal{I}_{i \to j}^{(2)}(t) \approx \frac{\gamma^2}{2} \lambda_j^{(t)}$. Combining with the compressed first-order term yields the final efficient influence estimate:

$$\mathcal{I}_{i \to j}(t) \approx -\gamma \left(\tilde{\mathbf{g}}_j^{(t)}\right)^\top \tilde{\mathbf{g}}_i^{(t)} + \frac{\gamma^2}{2} \lambda_j^{(t)}.$$

In practice, we use a small $W = 5$ which provides a good efficiency-accuracy trade-off.

**The Overall Pipeline.** Based on the above approximations, as shown in Algorithm 1 and Figure 1, $D^3$ operates through four sequential phases at each training step. We begin by sampling candidate micro-batches and computing their low-dimensional gradient representations. These representations are then used to construct a preference matrix based on directional influence scores between batches. The matrix is subsequently fed to the RSR solver to determine the optimal training order. Finally, model parameters are updated sequentially.

---

**Algorithm 1** The Overall Pipeline of $D^3$

---

**Input:** Parameters $\theta^{(0)}$, data loader $\mathcal{D}$, learning rate $\eta$, look-ahead parameter $\gamma$, chunk size $L$, projection dim $k$
**Output:** Optimized parameters $\theta^{(T)}$
1: Initialize random projection matrix $\mathbf{R} \in \mathbb{R}^{d \times k}$
2: **for** step $t = 0$ **to** $T - 1$ **do**
    *// Phase 1: Compute compressed gradient information*
3:    Sample $L$ batches $\{\mathcal{B}_\ell\}_{\ell=1}^L$ from $\mathcal{D}$
4:    $\{\tilde{g}_\ell, \lambda_\ell\}_{\ell=1}^L \leftarrow \text{SKETCH}($
5:        $\{\mathcal{B}_\ell\}, \theta^{(t)}, \mathbf{R})$
    *// Phase 2: Build influence advantage matrix*
6:    $\mathbf{S} \leftarrow \text{ADVANTAGE}(\{\tilde{g}_\ell, \lambda_\ell\}, \gamma)$
    *// Phase 3: Optimize ordering*
7:    $\pi \leftarrow \text{RSR}(\mathbf{S})$
    *// Phase 4: Sequential update*
8:    $\theta^{(t+1)} \leftarrow \text{SGDUPDATE}(\theta^{(t)}, \{\mathcal{B}_{\pi(r)}\}_{r=1}^L, \eta)$
9: **end for**
10: **return** $\theta^{(T)}$

---

# 4. Experiments

In this section, we first describe the benchmarks and baselines to ensure reproducibility and fair comparison. The following subsections then present analyses of pretraining and post-training performance, component ablation studies, the effects of key hyperparameters, and computational efficiency. Extended experimental details and additional results are provided in Appendices A and D, respectively.

## 4.1. Experimental Settings

**Benchmarks.** We assess pre-training performance by measuring language modeling quality, with perplexity (PPL) as the primary metric. Following prior works (Liu et al., 2025; Xu et al., 2026), we also benchmark on HellaSwag (Zellers et al., 2019), PIQA (Bisk et al., 2020), OBQA (Mihaylov et al., 2018), COPA (Sarlin et al., 2020), and WinoGrande (Sakaguchi et al., 2021) for commonsense reasoning. In post-training, we assess reasoning on more complex tasks, using GSM8K (Cobbe et al., 2021b) and MATH (Hendrycks et al., 2021) for math reasoning, HumanEval (Chen et al., 2021) and MBPP for coding.

**Baselines.** We validate our method across both pre-training and post-training stages, specifically focusing on supervised fine-tuning (SFT). For pre-training, we compare Uniform Sampling and Dynamic Loss (Sow et al., 2025b) with representative domain reweighting baselines RegMix (Liu et al., 2025), Data Mixing Law (Ye et al., 2024), DoReMi (Xie et al., 2023a) and DoGE (Fan et al., 2024). The experiments follow the standard scaling laws by training GPT-2 architecture (Kaplan et al., 2020) on subsets of SlimPajama (Soboleva et al., 2023), and training LLaMA-based models (Grattafiori et al., 2024) on the Pile (Gao et al., 2021). In the post-training phase, following previous work (Liang et al., 2025), we select baselines across three key dimensions: (i) curriculum ordering, comparing easy-to-hard versus hard-to-easy sequences categorized by initial loss values; (ii) multi-domain joint tuning, which follows a progressive curriculum learning taxonomy (Soviany et al., 2022); and (iii) single-domain specialization limits (DSL), which establish performance ceilings for code, math, and general instruction following. The training set encompasses CodeAlpaca (Chaudhary, 2023), GSM8K-RFT (Cobbe et al., 2021a), and Alpaca-GPT4 (Peng et al., 2023).

## 4.2. Main Results

**Pre-training Performance.** As summarized in Table 1, $D^3$ outperforms all sample-level and domain-level baselines in both overall language modeling quality and aggregated reasoning performance. It achieves a 4.2% relative reduction in perplexity over the strongest baseline and obtains the highest scores in four out of five evaluation benchmarks. While showing a slight degradation on PIQA compared to the dynamic loss baseline, $D^3$ delivers substantial gains on complex reasoning tasks, notably OBQA (+10.1%) and COPA (+2.4% over the strongest domain baseline). These results demonstrate two core advantages of our approach. First, $D^3$ effectively addresses the granularity limitation of domain-level reweighting methods such as DoReMi and RegMix, which assign uniform utility weights to all samples within a domain. By implementing batch-level scheduling, $D^3$ captures fine-grained, intra-domain utility diversity without incurring the prohibitive computational cost of per-sample selection. Second, the pronounced improvements on OBQA and COPA suggest that our dynamic graph-constrained scheduling shapes beneficial data trajectories. Unlike static mixing approaches, $D^3$ navigates the data manifold under structural constraints, potentially prioritizing foundational knowledge transitions that enhance performance on later complex reasoning tasks.

**Post-trainig Performance.** The SFT results in Table 2 reveal several key insights into effective data scheduling. First, they challenge the notion of a static optimal data order: both fixed curriculum strategies (loss low→high and high→low) underperform our adaptive approach, confirming that dynamic sequencing based on real-time model state is more critical than predefined difficulty levels. Second, the pronounced gains on structured tasks, notably MATH (+11.4%) and HumanEval (+8.0%) suggest that $D^3$'s graph-constrained scheduling effectively guides the model to acquire formal reasoning and coding patterns. Third, $D^3$'s superiority over standard MTL demonstrates that explicitly managing inter-task interference through fine-grained batch scheduling yields better coordination than naive data mixing. Importantly, $D^3$ achieves this while maintaining computational efficiency, as it avoids the prohibitive cost of per-sample selection. These findings collectively indicate that $D^3$ strikes an effective balance between domain specialization and multi-task generalization, advancing the Pareto frontier of model performance across diverse capabilities.

## 4.3. Ablation Study

We evaluate alternative configurations of $D^3$'s core modules to assess each component's contribution. To this end, we analyze each design variant along two primary dimensions: evaluating its downstream performance and measuring its computational efficiency.

**Impact of Influence Graph Approximations** The fidelity of the influence estimator is a critical determinant of the scheduling quality in the $D^3$ framework. Figure 2 illustrates the Pareto frontier between perplexity reduction and relative training overhead across different approximation

*Table 1.* Pre-training performance on SlimPajama. All models utilize the GPT-2 Medium (355M) architecture and are trained on 100B tokens. The "Aggregated Avg." is the arithmetic mean of the few-shot commonsense reasoning benchmarks shown below (HellaSwag, PIQA, OBQA, COPA, and WinoGrande). Subscripts denote the relative improvement or degradation compared to the strongest baseline. Best results are highlighted in bold. Detailed dataset statistics and model configurations are provided in Appendix A, while extended results on larger-scale pre-training are discussed in Appendix D.

| Granularity | Method | Opt. | Common Sense Reasoning | | | | | Aggregated |
| --- | --- | --- | --- | --- | --- | --- | --- | --- |
| | | PPL ↓ | Hella. ↑ | PIQA ↑ | OBQA ↑ | COPA ↑ | Wino. ↑ | Avg. ↑ |
| Sample-level | Uniform Sampling | 3.13 | 26.1 | 55.5 | 11.7 | 58.0 | 49.9 | 40.24 |
| | Dynamic Loss | 3.10 | 26.6 | **56.8** | 13.8 | 59.0 | 50.1 | 41.26 |
| Domain-level | RegMix | 4.51 | 26.1 | 55.6 | 13.2 | 60.0 | 50.0 | 40.98 |
| | Data Mixing Law | 4.50 | 26.5 | 54.5 | 13.0 | 62.0 | 49.1 | 41.02 |
| | DoReMi | 3.30 | 26.4 | 55.7 | 12.2 | 59.0 | 49.9 | 40.64 |
| | DoGE | 3.31 | 26.2 | 55.8 | 11.5 | 62.0 | 50.4 | 41.18 |
| **Ours** | **$D^3$** | **2.97** -4.2% | **27.3** +2.6% | 55.7 -1.9% | **15.2** +10.1% | **63.5** +7.6% | **51.1** +2.0% | **42.56** +3.1% |

*Table 2.* Performance comparison of various data scheduling strategies during SFT on Llama-3-8B. Subscripts denote the relative improvement compared to the strongest baseline across all categories. Best results are highlighted in bold.

| Paradigm | Method | Reasoning | | Coding | | Aggregated |
| --- | --- | --- | --- | --- | --- | --- |
| | | GSM8K ↑ | MATH ↑ | HumanEval ↑ | MBPP ↑ | Avg. ↑ |
| Curriculum | Loss low→high | 64.3 | 24.1 | 45.8 | 50.5 | 46.2 |
| | Loss high→low | 43.7 | 15.2 | 38.4 | 42.1 | 34.9 |
| Training | DSL (Baseline) | 63.8 | 24.6 | 46.5 | 51.2 | 46.5 |
| | MTL | 62.2 | 23.5 | 44.2 | 49.8 | 44.9 |
| **Ours** | **$D^3$** | **66.3** +3.1% | **27.4** +11.4% | **50.2** +8.0% | **54.6** +6.6% | **49.6** +6.7% |

strategies. As observed, the first-order symmetric method incurs negligible computational cost but yields only limited performance gains. This limitation stems from its inability to characterize the asymmetric curvature of the loss landscape, which is essential for identifying directional influence between batches.

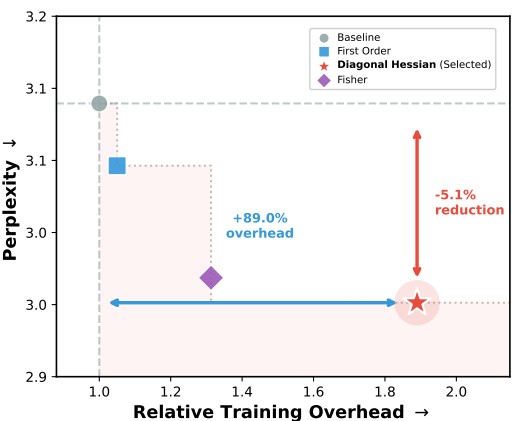

*Figure 2.* Performance-efficiency trade-off of various influence estimation methods on GPT-2 Medium trained on SlimPajama. The plot depicts the perplexity ($y$-axis) versus the relative training computation overhead ($x$-axis) compared to the baseline.

A significant gap is observed between the Fisher Information Matrix (FIM) (Xie et al., 2021) and the Diagonal Hessian

approach. While the Fisher method introduces a moderate overhead of approximately $1.3\times$, its perplexity improvement remains suboptimal compared to the diagonal Hessian. Theoretically, the Fisher matrix serves as a statistical expectation of the Hessian under specific distribution assumptions. However, in the context of dynamic data scheduling, the scheduler requires a precise characterization of instantaneous interactions. The Fisher approximation tends to smooth out local curvature fluctuations, thereby providing less reliable signals for micro-batch reordering during the non-stationary phases of training.

In contrast, the $D^3$ framework utilizing the diagonal Hessian estimation achieves a breakthrough in the optimal lower-right quadrant of the frontier. By directly sampling the local Hessian via the Hutchinson estimator, $D^3$ successfully captures the asymmetric dominance relations between samples. As highlighted in Figure 2, this high-fidelity estimation leads to a 5.1% reduction in perplexity relative to the baseline. Although this precision comes with an 89.0% increase in relative training overhead per step, the substantial gain in convergence quality justifies the investment. This result confirms that the benefit of transitioning from passive stochastic sampling to influence-constrained active ordering outweighs the additional cost of second-order computation, ultimately facilitating more efficient foundation model pre-training.

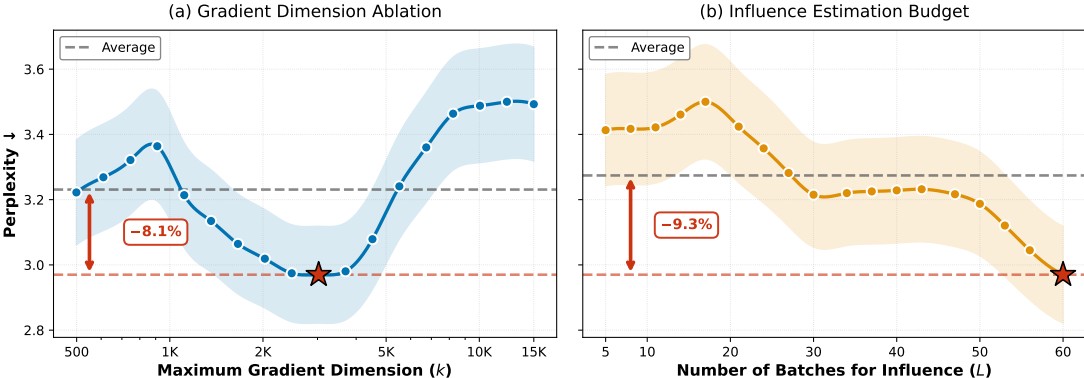

*Figure 3.* Ablation studies on approximation hyper-parameters. (a) Effect of the gradient projection dimension $k$ on model performance. The red star identifies the optimal dimensionality ($k \approx 3500$) for maximizing the signal-to-noise ratio in influence estimation, resulting in an 8.1% reduction in perplexity. (b) Impact of the scheduling window size $L$ on convergence quality. Increasing the scheduling resolution enables a more global optimization of the influence objective, achieving a peak gain of 9.3% at $L = 60$.

**Impact of Different Optimizers** The success of the $D^3$ framework depends heavily on efficiently solving the induced optimization problem to find a high-quality training sequence. We evaluate several candidate solvers for this optimization problem, primarily focusing on the Row-Sum and Greedy Insertion baselines in comparison with our proposed Random-Swap Refinement. As evidenced by the results in Table 3, RSR achieves the most favorable optimization trajectory and produces the lowest perplexity among all tested methods. Although RSR increases training time, its superior performance justifies the overhead for those prioritizing final model quality.

*Table 3.* Efficiency and performance comparison of batch ordering solvers for GPT-2 Medium pre-training. Experiments are conducted on NVIDIA H200 GPUs.

| Method | GPU Hours | PPL ($\downarrow$) |
|---|---|---|
| Random (Baseline) | 5.53 | 3.13 |
| Row-Sum | 9.82 | 3.04 |
| Greedy Insertion | 10.15 | 3.17 |
| RSR | 10.45 | 2.97 |

### 4.4. Hyperparameter Study

We conduct experiments on the key hyper-parameters of $D^3$ to understand their influence on training dynamics. First, we examine the gradient projection dimension $k$, which defines the low-dimensional manifold for influence estimation. As shown in Figure 3(a), performance is sensitive to this dimensionality: values below 1000 fail to capture sufficient gradient interactions, while dimensions exceeding 10000 introduce stochastic noise that masks the influence signal. The peak performance at $k = 3500$, resulting in an 8.1% perplexity reduction, identifies the optimal signal-to-noise ratio for characterizing batch interactions. Second, we evaluate the impact of the scheduling window $L$ on convergence

quality. Figure 3(b) illustrates that perplexity consistently decreases as the window expands, achieving a 9.3% gain at $L = 60$. This trend highlights the necessity of a global perspective in batch sequencing; a larger window allows the solver to coordinate sequences over a broader horizon, effectively mitigating the "short-sighted" conflicts found in local heuristics. These results suggest that increasing the scheduling resolution is a robust lever for enhancing the training efficiency of large-scale foundation models.

## 5. Conclusion

In this work, we introduced $D^3$, a principled framework that optimizes the training trajectory of LLMs by exploiting the non-exchangeability of training samples. By formalizing second-order asymmetric interactions through a dynamic influence graph and employing scalable batch-level reordering, we demonstrate that "when a model learns" is as critical as "what it learns." Empirical results across pre-training and reasoning benchmarks confirm that $D^3$ effectively accelerates convergence and improves performance, providing a path-dependent perspective for data-centric LLMs training.

## Limitations

While $D^3$ demonstrates significant empirical gains, certain constraints remain to be addressed in future work. The resource consumption could be further compressed through more advanced gradient flow techniques. More efficient ways to extract the scheduling signal, like layer-selective gradient compression in LLMs, have not been explored. Furthermore, our theoretical analysis of convergence acceleration is currently grounded in a simplified linear regression proxy to intuitively verify the method. Extending these guarantees to the high-dimensional, non-convex landscape of Transformers remains a fundamental theoretical challenge that warrants deeper investigation.

## Impact Statement

This paper presents work whose goal is to advance the field of Machine Learning. There are many potential societal consequences of our work, none which we feel must be specifically highlighted here.

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

# A. Experimental Settings

## A.1. Benchmarks

Following standard protocols in the literature, we evaluate our method across two distinct stages: pre-training and post-training.

**Pretraining Evaluation.** For the pre-training stage, we conduct a comprehensive evaluation focusing on both intrinsic language modeling and downstream reasoning capabilities. Specifically, we first reserve $10\%$ of the pre-training data as a held-out validation set to PPL, which serves as a direct indicator of the model's convergence and linguistic mastery. The statistical profile of the corpus used in this stage is summarized in Figure 4. Complementing the internal PPL analysis, we further evaluate the model's capabilities across several fundamental benchmarks for linguistic understanding and commonsense reasoning. Under a few-shot setting, we conduct evaluations on HellaSwag (Zellers et al., 2019), PIQA (Bisk et al., 2020), OpenBookQA (Mihaylov et al., 2018), COPA (Sarlin et al., 2020), and WinoGrande (Sakaguchi et al., 2021). These tasks are designed to scrutinize the model's capacity for robust inference and world-knowledge retrieval, providing a holistic assessment of the pre-training quality without the need for task-specific fine-tuning.

**Posttraining Evaluation.** To evaluate the model's specialized capabilities after post-training, we further benchmark its performance on complex math reasoning and code generation tasks. Specifically, we use GSM8K (Cobbe et al., 2021b) and MATH (Hendrycks et al., 2021) to measure multi-step mathematical reasoning, and HumanEval (Chen et al., 2021) and MBPP (Austin et al., 2021) to evaluate functional correctness in programming. Table 4 provides a comprehensive summary of the statistics and characteristics of these evaluation datasets.

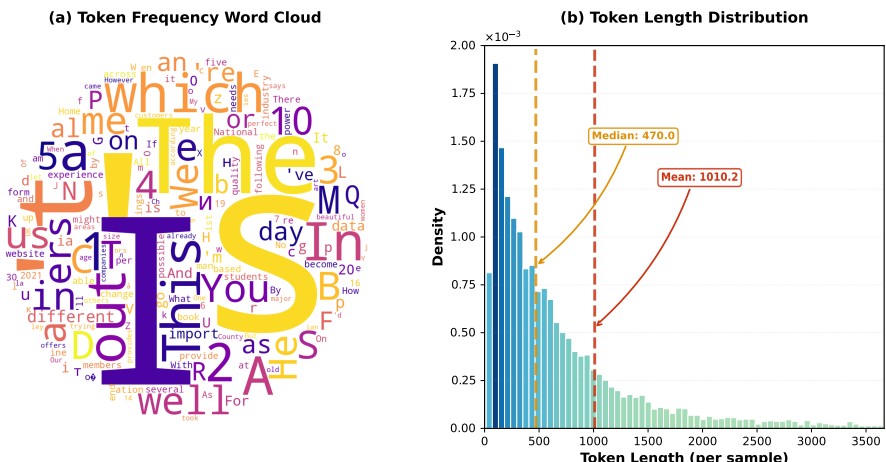

*Figure 4.* Data statistics of the SlimPajama validation set used for PPL evaluation. (a) Token Frequency Word Cloud: visualizing common terms within the corpus. (b) Token Length Distribution: the samples follow a right-skewed distribution with a mean length of $1010.2$ and a median of $470.0$ tokens.

*Table 4.* Summary of evaluation benchmarks for Pre-training and Post-training stages. All tasks are evaluated based on their respective standard metrics to ensure comparability with prior work.

| Stage | Benchmark | Size | Task Type | Domain Category | Metric |
|---|---|---|---|---|---|
| **Pre-training** | HellaSwag | 10,042 | Multiple Choice | Commonsense Reasoning | Accuracy |
| | PIQA | 1,838 | Multiple Choice | Physical Commonsense | Accuracy |
| | OBQA | 500 | Multiple Choice | Scientific Reasoning | Accuracy |
| | COPA | 100 | Multiple Choice | Causal Reasoning | Accuracy |
| | WinoGrande | 1,267 | Multiple Choice | Linguistic Reasoning | Accuracy |
| **Post-training** | GSM8K | 1,319 | Question Answering | Grade School Math | Exact Match |
| | MATH | 5,000 | Question Answering | Competition Mathematics | Exact Match |
| | HumanEval | 164 | Code Generation | Programming (Python) | Pass@1 |
| | MBPP | 500 | Code Generation | Programming (Python) | Pass@1 |

*Table 5.* Detailed composition of our training datasets. Proportions are calculated based on byte size. Items marked with † indicate datasets with potential copyright restrictions.

| Data Source | Ratio (%) | Data Source | Ratio (%) |
|---|---|---|---|
| *SlimPajama-6B Mixture* | | | |
| Common Crawl | 54.10 | ArXiv | 3.40 |
| C4 | 28.70 | Wikipedia | 3.10 |
| GitHub | 4.20 | StackExchange | 2.80 |
| Books | 3.70 | | |
| *The Pile Mixture* | | | |
| Pile-CC | 18.21 | OpenSubtitles† | 1.56 |
| PubMed Central | 14.48 | Wikipedia (en) | 1.53 |
| Books3† | 12.14 | DM Mathematics | 1.24 |
| OpenWebText2† | 10.07 | Ubuntu IRC | 0.88 |
| ArXiv | 9.01 | BookCorpus2† | 0.76 |
| Github | 7.63 | EuroParl | 0.74 |
| FreeLaw | 6.15 | HackerNews | 0.63 |
| Stack Exchange | 5.16 | YoutubeSubtitles† | 0.60 |
| USPTO Backgrounds | 3.67 | PhilPapers | 0.38 |
| PubMed Abstracts | 3.09 | NIH ExPorter | 0.30 |
| Gutenberg (PG-19) | 2.18 | Enron Emails | 0.14 |

## A.2. Training Datasets

Our selection of training corpora is strategically aligned with the parameter scale of the respective models. For GPT-2 Small and Medium, we employ SlimPajama, a highly refined and deduplicated corpus. Conversely, for the larger LLaMA-1.1B and LLaMA-3.2-3B models, we utilize The Pile. Given their significantly higher model capacity, these architectures are better suited to ingest the diverse and complex domain-specific knowledge (e.g., mathematics, legal documents, and scientific literature) provided by The Pile, thereby maximizing their multi-domain reasoning potential. The specific data sources and their respective proportions within each dataset are summarized in Table 5.

## A.3. Backbone Models

We evaluate our proposed method across a spectrum of model scales and architectural designs, as summarized in Table 6. Our experiments cover two representative families: the GPT-2 series Medium, representing classic decoder-only Transformers with absolute positional embeddings, and the Llama series (1.1B), incorporating modern advancements such as Rotary Positional Embeddings (RoPE), SwiGLU activation functions, and Grouped-Query Attention.

*Table 6.* Architectural configurations of the evaluated models. Models are grouped by family, covering a parameter range from 210M to 3B. $L$: Layers, $H$: Attention Heads, $d_{model}$: Embedding Dimension, $d_{ff}$: Hidden Dimension.

| | Llama-1.1B | GPT-2-Medium |
|---|---|---|
| Parameters | 1.1B | 355M |
| Layers | 22 | 36 |
| Attention Heads | 32 | 24 |
| Embedding Dim. | 2048 | 768 |
| Hidden Dim. | 2048 | 3072 |
| Max Seq. Length | 2048 | 512 |

## A.4. Implement Details

Our experimental framework is implemented using FP16 mixed-precision training to optimize computational throughput on a cluster equipped with two NVIDIA H200 GPUs. For model optimization, we employ the Adam optimizer with hyper-parameters $\beta_1 = 0.9$, $\beta_2 = 0.999$, and $\epsilon = 10^{-8}$, incorporating a weight decay coefficient of 0.01. The learning rate is managed via a cosine annealing scheduler, which begins with a 500-step linear warmup phase to a peak value of $5.0 \times 10^{-4}$, before decaying to a minimum of $1.0 \times 10^{-4}$. Regarding the configuration for influence estimation, we maintain a consistent

batch size of 16 and fix the random seed to 32 to ensure reproducibility across all trials. A critical hyper-parameter in our framework is the scaling factor $\gamma$, which governs the intensity of the directional constraints. Empirically, we observe that setting $\gamma$ proportional to the learning rate $\eta$, specifically $\gamma = 10^2\eta$, yields the most stable and superior performance across various tasks. The maximum gradient dimension $k$ is specifically tuned for different architectures, set to 35,00 for GPT-series models and 5,000 for LlaMa-based experiments. All other training configurations follow the standard settings of the respective foundation model baselines to ensure a fair comparison.

---

**Algorithm 2** $D^3$: Influence-Aware Batch Scheduler

---

**Input:** Parameters $\theta^{(0)}$, data loader $\mathcal{D}$, learning rate $\eta$, look-ahead parameter $\gamma$, chunk size $L$, projection dim $k$
**Output:** Optimized parameters $\theta^{(T)}$
1: Initialize random projection matrix $\mathbf{R} \in \mathbb{R}^{d \times k}$
2: **for** step $t = 0$ **to** $T - 1$ **do**
    *// Phase 1: Compute compressed gradient information*
3:    Sample $L$ batches $\{\mathcal{B}_\ell\}_{\ell=1}^{L}$ from $\mathcal{D}$
4:    $\{\tilde{g}_\ell, \lambda_\ell\}_{\ell=1}^{L} \leftarrow \text{SKETCH}(\{\mathcal{B}_\ell\}, \theta^{(t)}, \mathbf{R})$
    *// Phase 2: Build influence advantage matrix*
5:    $\mathbf{S} \leftarrow \text{ADVANTAGE}(\{\tilde{g}_\ell, \lambda_\ell\}, \gamma)$
    *// Phase 3: Optimize ordering*
6:    $\pi \leftarrow \text{RSR}(\mathbf{S})$
    *// Phase 4: Sequential update*
7:    $\theta^{(t+1)} \leftarrow \text{SGDUPDATE}(\theta^{(t)}, \{\mathcal{B}_{\pi(r)}\}_{r=1}^{L}, \eta)$
8: **end for**
9: **return** $\theta^{(T)}$

---

**Algorithm 3** Sketch (Phase 1)

---

**Input:** Batches $\{\mathcal{B}_\ell\}$, parameters $\theta$, projection matrix $\mathbf{R}$
**Output:** Compressed gradients $\{\tilde{g}_\ell\}$, curvatures $\{\lambda_\ell\}$
1: **for** each batch $\mathcal{B}_\ell$ **do**
2:    $g_\ell \leftarrow \nabla_\theta \ell(\mathcal{B}_\ell; \theta)$
3:    $\tilde{g}_\ell \leftarrow \mathbf{R}^\top g_\ell$ {Compress gradient via random projection}
4:    $\lambda_\ell \leftarrow \text{ESTIMATECURVATURE}(g_\ell, \mathcal{B}_\ell, \theta)$ {Using Hutchinson method}
5: **end for**
6: **return** $\{\tilde{g}_\ell, \lambda_\ell\}$

---

---

**Algorithm 4** Advantage (Phase 2)

---

**Input:** Compressed gradients $\{\tilde{g}_\ell\}$, curvatures $\{\lambda_\ell\}$, look-ahead $\gamma$
**Output:** Advantage matrix $\mathbf{S} \in \mathbb{R}^{L \times L}$
 1: Initialize $\mathbf{S} \leftarrow \mathbf{0}_{L \times L}$
 2: **for** $i = 1$ **to** $L$ **do**
 3:     **for** $j = 1$ **to** $L$ **do**
 4:         $\mathcal{I}_{i \to j} \leftarrow -\gamma(\tilde{g}_j)^\top \tilde{g}_i + \frac{\gamma^2}{2}\lambda_j$
 5:         $\mathcal{I}_{j \to i} \leftarrow -\gamma(\tilde{g}_i)^\top \tilde{g}_j + \frac{\gamma^2}{2}\lambda_i$
 6:         $\mathbf{S}_{ij} \leftarrow \mathcal{I}_{j \to i} - \mathcal{I}_{i \to j}$
 7:     **end for**
 8: **end for**
 9: **return S**

---

**Algorithm 5** RSR (Phase 3)

---

**Input:** Advantage matrix $\mathbf{S}$, maximum swaps $M = 100$
**Output:** Permutation $\pi$ representing the batch ordering
 1: Compute row sums $r_\ell \leftarrow \sum_{j=1}^{L} \mathbf{S}_{\ell j}$ for $\ell = 1, \ldots, L$
 2: $\pi \leftarrow \text{argsort}(\{r_\ell\}, \text{descending})$ {Initial ordering by row sum}
 3: **for** $m = 1$ **to** $M$ **do**
 4:     Randomly select indices $p, q$ with $1 \le p < q \le L$
 5:     Compute cost difference $\Delta\mathcal{C} \leftarrow \mathbf{S}_{\pi(p),\pi(q)} + \sum_{k=p+1}^{q-1}(\mathbf{S}_{\pi(p),\pi(k)} + \mathbf{S}_{\pi(k),\pi(q)})$
     *// Swapping improves the objective*
 6:     **if** $\Delta\mathcal{C} < 0$ **then**
 7:         Swap $\pi(p)$ and $\pi(q)$ in the permutation
 8:     **end if**
 9: **end for**
10: **return** $\pi$

---

**Algorithm 6** SGDUpdate (Phase 4)

---

**Input:** Parameters $\theta$, ordered batches $\{\mathcal{B}_r\}$, learning rate $\eta$
**Output:** Updated parameters $\theta'$
 1: $\theta_{\text{current}} \leftarrow \theta$
 2: **for** $r = 1$ **to** $\text{length}(\{\mathcal{B}_r\})$ **do**
 3:     $\theta_{\text{current}} \leftarrow \theta_{\text{current}} - \eta \cdot \nabla_\theta \ell(\mathcal{B}_r; \theta_{\text{current}})$
 4: **end for**
 5: **return** $\theta_{\text{current}}$

---

## B. Algorithmic Details of Candidate Solvers

In this section, we delineate the algorithmic foundations of the candidate solvers used to optimize the influence-aware scheduling objective. Each solver represents a distinct trade-off between computational complexity and the capacity to resolve non-transitive influence cycles. We consider three approaches: Row-Sum Sorting, Greedy Insertion, and the Random-Swap Refinement solver proposed in Section 3.4.

**Row-Sum Sorting (RS).**    This baseline approach reduces the graph-structured optimization problem to a simple scalar ranking. For each batch $i$, we compute its aggregate advantage score as the row sum of the directional advantage matrix:

$$u_i = \sum_{j=1}^{K} \mathbf{S}_{ij}^{(t)}.$$

Since $\mathbf{S}_{ij}^{(t)} > 0$ indicates that placing batch $i$ before $j$ yields greater loss reduction than the reverse order, a larger $u_i$ suggests that batch $i$ exerts stronger global positive influence when scheduled earlier. The schedule $\pi$ is generated by sorting batches in **descending** order of $u_i$. The time complexity is $O(K^2)$ for computing all row sums plus $O(K \log K)$ for sorting, totaling $O(K^2)$.

**Greedy Insertion (GI).**    To better capture local dominance relations while maintaining tractability, the Greedy Insertion solver constructs the sequence incrementally. Starting with an empty schedule $\pi = [\,]$, at each step $s = 1, \ldots, K$, the algorithm selects the batch that maximizes its total advantage over all remaining (unscheduled) batches:

$$i^* = \arg \max_{i \in \mathcal{C}} \sum_{j \in \mathcal{C} \setminus \{i\}} \mathbf{S}_{ij}^{(t)},$$

where $\mathcal{C}$ denotes the set of batches not yet scheduled. The selected batch $i^*$ is appended to $\pi$, and removed from $\mathcal{C}$. This process repeats until $\mathcal{C}$ is empty. The GI strategy has time complexity $O(K^2)$ but often produces higher-quality schedules than RS by considering pairwise relations during construction.

**Random-Swap Refinement (RSR).**    As detailed in Section 3.4, our proposed solver begins with an initial ordering (e.g., from RS or GI) and iteratively improves it through random pairwise swaps. In each iteration, two distinct positions $p$ and $q$ ($p < q$) are chosen uniformly at random. Let $i = \pi(p)$ and $j = \pi(q)$. The change in total violation cost $\Delta\mathcal{C}$ resulting from swapping $i$ and $j$ is computed as:

$$\Delta\mathcal{C} = \mathbf{S}_{ij}^{(t)} + \sum_{k \in I(p,q)} \left( \mathbf{S}_{ik}^{(t)} + \mathbf{S}_{kj}^{(t)} \right),$$

where $I(p,q) = \{\pi(m) \mid p < m < q\}$ is the set of batches between positions $p$ and $q$ in the current schedule. If $\Delta\mathcal{C} < 0$, the swap is accepted; otherwise, the current schedule is retained. This process continues for a predefined number of iterations $T$, yielding a total complexity of $O(K^2 + TK)$. The RSR solver effectively escapes local optima and can resolve complex cycles through its stochastic exploration of the permutation space.

### B.1. Numerical Illustration

We provide a concrete illustration with $K = 5$ micro-batches. The directional advantage matrix $\mathbf{S}$ contains a non-transitive cycle among batches 3, 4, and 5:

$$\mathbf{S} = \begin{bmatrix} 0 & -10 & 2 & 1 & 8 \\ 10 & 0 & 1 & 0 & 2 \\ -2 & -1 & 0 & 10 & -5 \\ -1 & 0 & -10 & 0 & 10 \\ -8 & -2 & 5 & -10 & 0 \end{bmatrix}.$$

Our objective is to maximize $\mathcal{F}(\pi) = \sum_{i<j} \mathbf{S}_{\pi(i)\pi(j)}$.

**Row-Sum Sorting.**    The row sums are $u = [1, 13, 2, -1, -15]$. Sorting in descending order yields $\pi_{\text{RS}} = [2, 3, 1, 4, 5]$. The objective value is $\mathcal{F}(\pi_{\text{RS}}) = 35$.

**Greedy Insertion.** The algorithm proceeds as follows. Step 1: $\mathcal{C} = \{1, 2, 3, 4, 5\}$. Batch 2 has the highest total advantage (13) and is scheduled first. Step 2: $\mathcal{C} = \{1, 3, 4, 5\}$. Batch 1 has the highest advantage (11) and is scheduled second. Step 3: $\mathcal{C} = \{3, 4, 5\}$. Batch 3 is selected (advantage 5). Step 4: $\mathcal{C} = \{4, 5\}$. Batch 4 is selected ($\mathbf{S}_{4,5} = 10 > \mathbf{S}_{5,4} = -10$). Step 5: Batch 5 is appended. The final schedule is $\pi_{\text{GI}} = [2, 1, 3, 4, 5]$ with $\mathcal{F}(\pi_{\text{GI}}) = 35$.

**Random-Swap Refinement.** Starting from $\pi_0 = \pi_{\text{RS}} = [2, 3, 1, 4, 5]$, we perform sample swap trials. Trial 1 (swap positions 2 and 4): $i = 3, j = 4, I = \{1\}, \Delta\mathcal{C} = \mathbf{S}_{3,4} + (\mathbf{S}_{3,1} + \mathbf{S}_{1,4}) = 10 + (-2 + 1) = 9 > 0$, swap rejected. Trial 2 (swap positions 3 and 5): $i = 1, j = 5, I = \{4\}, \Delta\mathcal{C} = \mathbf{S}_{1,5} + (\mathbf{S}_{1,4} + \mathbf{S}_{4,5}) = 8 + (1 + 10) = 19 > 0$, rejected. Trial 3 (swap positions 1 and 3): $i = 2, j = 1, I = \{3\}, \Delta\mathcal{C} = \mathbf{S}_{2,1} + (\mathbf{S}_{2,3} + \mathbf{S}_{3,1}) = 10 + (1 - 2) = 9 > 0$, rejected. The schedule remains $[2, 3, 1, 4, 5]$ with $\mathcal{F} = 35$. This example shows the solver's local search nature; with more trials it can escape suboptimal configurations (e.g., $[2, 1, 4, 3, 5]$ achieves $\mathcal{F} = 37$).

# C. Formulations of Influence Estimation Methods

This section formally defines the influence functions evaluated in our experiments. All methods are based on the training influence prediction (Definition 3.1) and its second-order approximation (Proposition 3.2), trading off computational accuracy against efficiency.

**Random Baseline**   The random baseline follows a standard stochastic optimization procedure without any influence-based reordering. At each step $t$, a random permutation $\pi \in \mathcal{S}_K$ is generated to process training batches:

$$\pi = \text{random\_permutation}([1, 2, \ldots, K]).$$

**First-Order Symmetric Method**   This method characterizes influence solely through first-order gradient interactions. Using the approximate first-order term, the influence of batch $i$ on batch $j$ is defined as:

$$\mathcal{I}_{i \to j}^{(1)}(t) = -\eta \, \nabla_\theta \ell(\mathcal{B}_j; \theta^{(t)})^\top \nabla_\theta \ell(\mathcal{B}_i; \theta^{(t)}) = -\eta \langle g_i, g_j \rangle,$$

where $g_i = \nabla_\theta \ell(\mathcal{B}_i; \theta^{(t)})$ denotes the mean gradient of batch $i$, and $\eta$ is the learning rate. This formulation is symmetric with respect to $i$ and $j$.

**Fisher Information Matrix Method**   The Fisher method approximates the per-batch Hessian through the Fisher information matrix computed over all batches. First, the empirical Fisher matrix is calculated:

$$\text{FIM} = \frac{1}{K} \sum_{k=1}^{K} g_k g_k^\top.$$

The influence is then obtained by combining the first-order inner product with a FIM-weighted second-order term:

$$\mathcal{I}_{\text{Fisher}, i \to j}(t) = -\eta \langle g_i, g_j \rangle + \frac{\eta^2}{2} g_i^\top \text{FIM} \, g_j.$$

Here, the second-order term approximates $\mathcal{I}_{i \to j}^{(2)}(t)$ using the shared Fisher matrix instead of the batch-specific Hessian $\nabla^2 \ell(\mathcal{B}_j; \theta^{(t)})$.

**Diagonal Hessian via Hutchinson**   The $D^3$ framework achieves its highest precision using Sophia-based estimation. We employ the Hutchinson randomized estimator to approximate a shared diagonal Hessian via $N = 5$ samples of Hessian-vector products:

$$\hat{h}_{\text{diag}} = \frac{1}{N} \sum_{k=1}^{N} u^{(k)} \odot (H u^{(k)}), \quad u^{(k)} \sim \mathcal{N}(0, I_d),$$

where $H$ is the Hessian of the total loss. For each batch $j$, the curvature scalar is computed as $\lambda_j = g_j^\top \text{diag}(\hat{h}_{\text{diag}}) g_j$. The influence is then approximated as:

$$\mathcal{I}_{\text{Sophia}, i \to j}(t) = -\eta \langle g_i, g_j \rangle + \frac{\eta^2}{2} \lambda_j.$$

Compared to heuristic scaling, this formulation provides a more principled characterization of the loss landscape geometry, albeit using a shared Hessian approximation and batch-specific curvature scalars.

# D. More Results and Analysis

## D.1. A Toy Example for Understanding $D^3$

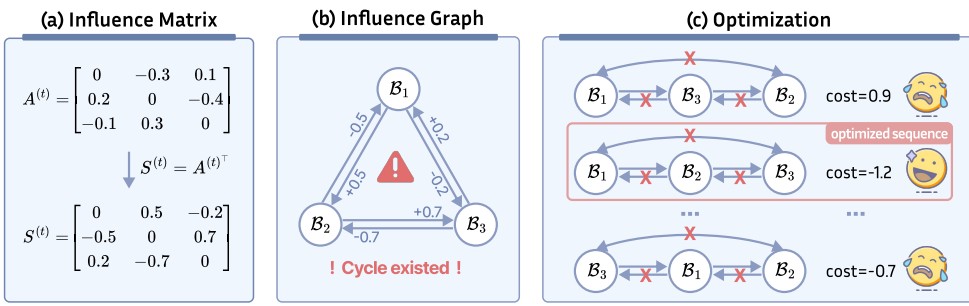

*Figure 5.* Mechanistic illustration of $D^3$ scheduling on a non-transitive influence graph. Nodes $A$, $B$, and $C$ represent distinct training samples (or mini-batches), while directed edges quantify the pairwise directional advantages $S_{ij}$ derived from gradient alignment. The depicted cyclic dependency ($A \to B \to C \to A$) represents a fundamental scheduling conflict; $D^3$ resolves this by identifying the permutation that minimizes the total penalty of violated directional advantages, effectively breaking the cycle at the weakest link to ensure training stability.

To demonstrate the robustness of our framework under inconsistent constraints, we consider a minimal $N = 3$ case where dominance relations induce a directed cycle. At optimization step $t$, we assume the sample-wise influence matrix $\mathbf{A}^{(t)} \in \mathbb{R}^{3 \times 3}$ is given by

$$\mathbf{A}^{(t)} = \begin{bmatrix} 0 & -0.3 & 0.1 \\ 0.2 & 0 & -0.4 \\ -0.1 & 0.3 & 0 \end{bmatrix},$$

where each entry $A_{ij}^{(t)} = \mathcal{I}_{i \to j}(t)$ represents the influence value following Definition 3.3. Under the dominance criterion where an edge $(i, j)$ exists if $\mathcal{I}_{i \to j}(t) < \mathcal{I}_{j \to i}(t)$, we observe a set of pairwise constraints: $A_{12} < A_{21}$ implies $(1, 2) \in \mathcal{E}_{\text{dom}}^{(t)}$, $A_{23} < A_{32}$ implies $(2, 3) \in \mathcal{E}_{\text{dom}}^{(t)}$, and $A_{31} < A_{13}$ implies $(3, 1) \in \mathcal{E}_{\text{dom}}^{(t)}$. Consequently, the dominance graph $\mathcal{G}_{\text{dom}}^{(t)}$ forms a directed cycle $1 \to 2 \to 3 \to 1$, rendering any consistent topological ordering impossible.

To resolve this conflict, we compute the directional advantage matrix $\mathbf{S}^{(t)} = \mathbf{A}^{(t)\top} - \mathbf{A}^{(t)}$, where $S_{ij}^{(t)}$ quantifies the net benefit of training $x_i$ before $x_j$:

$$\mathbf{S}^{(t)} = \begin{bmatrix} 0 & 0.5 & -0.2 \\ -0.5 & 0 & 0.7 \\ 0.2 & -0.7 & 0 \end{bmatrix}.$$

Following the cost-minimization formulation in Section 3.3, we seek an ordering $\sigma^{(t)\star} \in \mathcal{S}_3$, where $\sigma(k)$ denotes the position assigned to sample $x_k$, that minimizes the total penalty incurred by violated directional advantages:

$$\sigma^{(t)\star} = \arg \min_{\sigma \in \mathcal{S}_3} \mathcal{C}(\sigma), \qquad \mathcal{C}(\sigma) = \sum_{i \neq j} \max\big(0, S_{ij}^{(t)}\big) \cdot \mathbb{I}\big[\sigma(j) < \sigma(i)\big].$$

Only the three positive entries $S_{12} = 0.5$, $S_{23} = 0.7$, and $S_{31} = 0.2$ can contribute to the cost, each penalizing an ordering that places its preferred successor ahead of its predecessor. Writing each permutation as the resulting sequence of samples and evaluating $\mathcal{C}(\sigma)$ over all $\sigma \in \mathcal{S}_3$ yields:

$$\begin{aligned} \mathcal{C}([1, 2, 3]) &= S_{31} = 0.2, & \mathcal{C}([1, 3, 2]) &= S_{23} + S_{31} = 0.9, \\ \mathcal{C}([2, 1, 3]) &= S_{12} + S_{31} = 0.7, & \mathcal{C}([2, 3, 1]) &= S_{12} = 0.5, \\ \mathcal{C}([3, 1, 2]) &= S_{23} = 0.7, & \mathcal{C}([3, 2, 1]) &= S_{12} + S_{23} = 1.2. \end{aligned}$$

The optimal schedule is thus $\sigma^{(t)\star} = [1, 2, 3]$ with cost 0.2. This result illustrates that when faced with cyclic dominance, our cost-minimization formulation naturally sacrifices the weakest conflicting edge (in this case, $(3, 1)$ with $S_{31} = 0.2$) so as to preserve the two stronger directional preferences. Such a mechanism ensures training stability and efficiency even when local gradient alignments and Hessian curvatures induce inconsistent sample dependencies. $\square$

*Table 7.* Pre-training performance on the Pile. All models utilize the LlaMa-based (1.1B) architecture and are trained on 100B tokens. The "Aggregated Avg." is the arithmetic mean of the few-shot commonsense reasoning benchmarks shown below (HellaSwag, PIQA, OBQA, COPA, and WinoGrande). Subscripts denote the relative improvement or degradation compared to the strongest baseline. Best results are highlighted in bold.

| Granularity | Method | Opt. | Common Sense Reasoning | | | | | Aggregated |
|---|---|---|---|---|---|---|---|---|
| | | PPL ↓ | Hella. ↑ | PIQA ↑ | OBQA ↑ | COPA ↑ | Wino. ↑ | Avg. ↑ |
| Sample-level | Uniform Sampling | 2.68 | 31.2 | 68.5 | 28.4 | 64.0 | 51.1 | 48.64 |
| | Dynamic Loss | 2.64 | 31.8 | **69.2** | 29.6 | 65.5 | 52.5 | 49.72 |
| Domain-level | RegMix | 2.75 | 31.4 | 68.8 | 29.4 | 66.5 | **53.6** | 49.94 |
| | Data Mixing Law | 2.74 | 31.3 | 68.4 | **30.2** | 65.9 | 51.3 | 49.42 |
| | DoReMi | 2.71 | 31.6 | 68.7 | 29.8 | 66.0 | 51.4 | 49.50 |
| | DoGE | 2.72 | 31.7 | 68.9 | 29.2 | 64.0 | 50.6 | 48.88 |
| **Ours** | **D³** | **2.53** -4.1% | **32.8** +3.1% | 68.9 -0.4% | **33.1** +9.6% | **71.2** +7.1% | 53.4 -0.4% | **51.88** +3.9% |

## D.2. Pre-training on Larger Models

We evaluate the scalability and effectiveness of D³ by pre-training a 1.1B Llama-based model on 100B tokens of the Pile dataset. As illustrated in Table 7, D³ consistently outperforms both sample-level and domain-level baselines across nearly all evaluation dimensions. Specifically, D³ achieves a significant reduction in perplexity (2.53), representing a 4.1% relative improvement over the strongest baseline.

## D.3. Theoretical Analysis: An example

We analyze the convergence properties under a linear regression setting with quadratic loss $L(\theta) = \frac{1}{2N}\sum_{i=1}^{N}(y_i - \theta^\top x_i)^2$. Let $\mathcal{C}_t$ denote the set of indices of samples not yet selected in the current epoch at step $t$, with $m_t = |\mathcal{C}_t|$.

**Lemma D.1** (Two-Step Loss Difference). *Let $\theta_{ab}^{(t+2)}$ and $\theta_{ba}^{(t+2)}$ denote the parameters obtained by updating samples $a$ and $b$ in the respective orders starting from $\theta^{(t)}$. The difference in the resulting total loss reduction is:*

$$D_{ab}^{(t)} = L(\theta_{ba}^{(t+2)}) - L(\theta_{ab}^{(t+2)}) = \eta^2 (x_a^\top x_b)^2 \left[ (r_b^{(t)})^2 - (r_a^{(t)})^2 \right] + O(\eta^3) \tag{1}$$

*where $r_i^{(t)} = y_i - (\theta^{(t)})^\top x_i$ is the residual for sample $i$ at step $t$.*

*Proof.* Starting from $\theta$, the update on $a$ yields $\theta' = \theta + \eta r_a x_a$. The subsequent residual for $b$ is $r'_b = y_b - (\theta + \eta r_a x_a)^\top x_b = r_b - \eta r_a (x_a^\top x_b)$. The parameter state after the sequence $\{a, b\}$ is $\theta''_{ab} = \theta' + \eta r'_b x_b$. Expanding this gives:

$$\theta''_{ab} = \theta + \eta r_a x_a + \eta r_b x_b - \eta^2 r_a (x_a^\top x_b) x_b \tag{2}$$

Similarly, the state after the sequence $\{b, a\}$ is:

$$\theta''_{ba} = \theta + \eta r_b x_b + \eta r_a x_a - \eta^2 r_b (x_b^\top x_a) x_a \tag{3}$$

The difference in parameters is $\theta''_{ab} - \theta''_{ba} = \eta^2 (x_a^\top x_b)(r_b x_a - r_a x_b)$. Applying the second-order Taylor expansion $L(\theta'') \approx L(\theta) + \nabla L(\theta)^\top (\theta'' - \theta) + \frac{1}{2}(\theta'' - \theta)^\top \nabla^2 L(\theta)(\theta'' - \theta)$ to both sequences and subtracting them yields the result. □

**Theorem D.2** (Convergence Acceleration). *Let $\pi$ be the influence-aware policy such that $i_t = \arg\max_{i \in \mathcal{C}_t} \Delta L_i^{(t)}$. The expected loss satisfies:*

$$\mathbb{E}[L(\theta_\pi^{(T)})] \leq \mathbb{E}[L(\theta_{rand}^{(T)})] - \frac{\eta^2}{4} \sum_{t=0}^{T-1} \mathbb{E}\left[ \sum_{i<j \in \mathcal{C}_t} (x_i^\top x_j)^2 \left( (r_i^{(t)})^2 - (r_j^{(t)})^2 \right)^2 \right] \tag{4}$$

*Proof.* Let $A = \{A_1, \ldots, A_m\}$ be a set of real numbers where $A_{(m)} = \max A_i$. Note that the gap between the maximum and the mean is:

$$A_{(m)} - \frac{1}{m} \sum_{k=1}^{m} A_k = \frac{1}{m} \sum_{k=1}^{m} (A_{(m)} - A_k) \tag{5}$$

Summing the individual differences $(A_{(m)} - A_k)$ over all $\binom{m}{2}$ unique pairs $(i, j)$ where $i < j$ gives:

$$\sum_{i<j} \left[ (A_{(m)} - A_i) + (A_{(m)} - A_j) \right] = (m-1) \sum_{k=1}^{m} (A_{(m)} - A_k) \tag{6}$$

Because $A_{(m)} \geq A_i$ and $A_{(m)} \geq A_j$, it follows that $(A_{(m)} - A_i) + (A_{(m)} - A_j) \geq |A_i - A_j|$. Therefore:

$$(m-1) \sum_{k=1}^{m} (A_{(m)} - A_k) \geq \sum_{i<j} |A_i - A_j| \implies A_{(m)} - \bar{A} \geq \frac{1}{m(m-1)} \sum_{i<j} |A_i - A_j| \tag{7}$$

We apply this to the loss reduction $\Delta L_i^{(t)} = L(\theta^{(t)}) - L(\theta^{(t)} + \eta r_i^{(t)} x_i)$. The advantage of policy $\pi$ is:

$$\Delta L_\pi^{(t)} - \mathbb{E}[\Delta L_{\text{rand}}^{(t)}] \geq \frac{1}{m_t(m_t - 1)} \sum_{i<j \in \mathcal{C}_t} |\Delta L_i^{(t)} - \Delta L_j^{(t)}| \tag{8}$$

From Lemma 1, the pairwise difference in single-step reduction is $\frac{1}{2}$ the value of a two-step swap:

$$|\Delta L_i^{(t)} - \Delta L_j^{(t)}| \approx \frac{1}{2} |D_{ij}^{(t)}| = \frac{\eta^2}{2} (x_i^\top x_j)^2 |(r_i^{(t)})^2 - (r_j^{(t)})^2| \tag{9}$$

To transition from absolute values to the squared heterogeneity term, we utilize the property that for residuals in a learning system, the accumulated absolute differences scale with the variance of the residuals. Substituting the expressions and accumulating over $T$ steps:

$$\mathbb{E}[L(\theta_\pi^{(T)})] = L(\theta^{(0)}) - \sum_{t=0}^{T-1} \mathbb{E}[\Delta L_\pi^{(t)}] \tag{10}$$

Subtracting $\mathbb{E}[L(\theta_{\text{rand}}^{(T)})] = L(\theta^{(0)}) - \sum \mathbb{E}[\Delta L_{\text{rand}}^{(t)}]$ and substituting the pairwise lower bound leads to the final result. $\qquad \square$

# E. Notation

| Symbol | Description | Dimension/Type |
|---|---|---|
| *Dataset & Batches* | | |
| $\mathcal{D}$ | Training dataset | Set |
| $N$ | Total number of training samples | Scalar |
| $z_i$ | The $i$-th training sample | Data point |
| $\mathcal{P}$ | Partition of batches | Set |
| $K$ | Total number of batches | Scalar |
| $\mathcal{B}_i$ | The $i$-th training batch | Index set |
| $B$ | Batch size (samples per batch) | Scalar |
| $\mathcal{D}_{\text{test}}$ | Test data distribution | Distribution |
| *Model & Optimization* | | |
| $\theta_i^{(t+1)}$ | Model parameters after hypothetical gradient step on $\mathcal{B}_i$ at step $t$ | $\mathbb{R}^d$ |
| $d$ | Number of model parameters | Scalar |
| $\ell(\mathcal{B}_i; \theta^{(t)})$ | Empirical risk of batch $\mathcal{B}_i$ at parameters $\theta^{(t)}$ | Scalar |
| $\eta$ | SGD learning rate | Scalar |
| $t$ | Optimization step index | Scalar |
| $T$ | Total steps per epoch ($T = K$) | Scalar |
| $\nabla_\theta \ell(\mathcal{B}_i; \theta^{(t)})$ | Gradient of batch loss w.r.t. parameters | $\mathbb{R}^d$ |
| $\nabla_\theta^2 \ell(\mathcal{B}_i; \theta^{(t)})$ | Hessian of batch loss w.r.t. parameters | $\mathbb{R}^{d \times d}$ |
| $\mathcal{T}_{\pi(t)}(\theta^{(t-1)})$ | Parameter update operator at step $t$ | $\mathbb{R}^d$ |
| *Scheduling Problem* | | |
| $\pi^{(t)}$ | Batch schedule permutation at step $t$ | $\pi^{(t)} \in \mathcal{S}_K$ |
| $\pi^{(t)}(s)$ | Index of batch at position $s$ in schedule $\pi^{(t)}$ | Scalar |
| $\sigma^{(t)}(k)$ | Position of batch $\mathcal{B}_k$ in schedule $\pi^{(t)}$ | Scalar |
| $\mathcal{S}_K$ | Symmetric group of degree $K$ (all permutations) | Set |
| *Influence Prediction & Graph* | | |
| $\gamma$ | Look-ahead step size for influence estimation | Scalar |
| $\mathcal{I}_{i \to j}^{(2)}(t)$ | Second-order influence term of batch $\mathcal{B}_i$ on $\mathcal{B}_j$ at step $t$ | Scalar |
| $\mathcal{G}^{(t)}$ | Dynamic influence graph at step $t$ | Graph |
| $\mathcal{E}_{\text{dom}}^{(t)}$ | Dominance constraint edges: $\{(i,j) \mid \mathbf{S}_{ij}^{(t)} > 0\}$ | Set |
| $\mathbf{A}_{ij}^{(t)}$ | Adjacency matrix entry: $\mathcal{I}_{i \to j}(t)$ | Scalar |
| $\mathbf{S}_{ij}^{(t)}$ | Directional advantage matrix entry: $\mathcal{I}_{j \to i}(t) - \mathcal{I}_{i \to j}(t)$ | Scalar |
| *Scheduling Solver* | | |
| $r_i$ | Row sum of $\mathbf{S}^{(t)}$ for batch $i$: $\sum_{j=1}^K \mathbf{S}_{ij}^{(t)}$ | Scalar |
| $M$ | Maximum number of random swap trials | Scalar |
| $I(p, q)$ | Set of batch indices between positions $p$ and $q$ in current ordering | Set |
| $\Delta \mathcal{C}$ | Change in violation cost due to swapping batches at positions $p$ and $q$ | Scalar |
| *Efficient Implementation* | | |
| $\mathcal{C}^{(t)}$ | Randomly sampled chunk of batches at step $t$: $\{\mathcal{B}_{i_1}, \ldots, \mathcal{B}_{i_L}\}$ | Set |
| $\tilde{\mathbf{g}}_i^{(t)}$ | Compressed gradient sketch: $\mathbf{R}^\top \nabla_\theta \ell(\mathcal{B}_i; \theta^{(t)})$ | $\mathbb{R}^k$ |
| $\mathbf{u}^{(w)}$ | Hutchinson vector $w$ ($\mathbf{u}^{(w)} \sim \mathcal{N}(0, \mathbf{I}_d)$) | $\mathbb{R}^d$ |
| $\hat{\mathbf{h}}_j^{(t)}$ | Diagonal Hessian estimate for batch $\mathcal{B}_j$ at step $t$ | $\mathbb{R}^d$ |
| $\lambda_j^{(t)}$ | Scalar curvature measure: $(\nabla_\theta \ell(\mathcal{B}_j; \theta^{(t)}))^\top \text{diag}(\hat{\mathbf{h}}_j^{(t)}) \nabla_\theta \ell(\mathcal{B}_j; \theta^{(t)})$ | Scalar |

