# OpenReview forum: "D$^3$: Dynamic Directional Graph-Constrained Data Scheduling for LLM Training"
_ICML.cc/2026/Conference — ICML 2026 regular_

### Official Review · Reviewer_rGmr · 2026-03-12

**Soundness:** 2
**Presentation:** 2
**Significance:** 2
**Originality:** 3
**Overall Recommendation:** 4
**Confidence:** 3

**Summary:**

This paper proposes $D^3$, a data scheduling framework for LLM training. It formulates the complex interactions among training units (batches) as a dynamic influence graph. Subsequently, it frames the data scheduling problem as an optimization task and proposes a corresponding solver. To compare $D^3$ with existing data scheduling methods, the authors conduct experiments in both pre-training and post-training settings.

**Compliance With Llm Reviewing Policy:**

Affirmed.

**Final Justification:**

Most concerns are addressed except for the validation on larger models.

**Key Questions For Authors:**

1. Why is it best to choose first- and second-order influences in $D^3$?

2. Why do you only focus on SFT, and not RLVR, in LLM post-training?

3. The authors argue that $D^3$ can accelerate convergence. Can the authors show the training curve of $D^3$ as training units for one epoch and multiple epochs, compared with other baselines?

4. How effective is $D^3$ in a multiple-epoch training setting?

**Strengths And Weaknesses:**

**Strengths:**
1. Motivation: The motivation is sound, as data scheduling is an important issue in LLM training.

2. This paper proposes a novel perspective on the data scheduling problem in LLM training.

3. This paper conducts experiments on multiple benchmarks, including commonsense reasoning, mathematical reasoning, and coding.

**Weaknesses:**
1. Huge computational overhead for the claimed use case. The 89% increase in per-step training overhead (Figure 2) is extremely significant for LLM pre-training, where runs already consume millions of GPU-hours. The paper frames this as "manageable," but nearly doubling training cost per step is a serious practical limitation that undermines the core value proposition.

2. Limited scale of experiments. The primary experiments are conducted on GPT-2 Medium (355M) and LLaMA-1.1B, both on 100B tokens. By the standards of current LLM research, these are small-scale experiments. The SFT experiments use LLaMA-3-8B but with a relatively small training set (CodeAlpaca + GSM8K-RFT + Alpaca-GPT4). There is no evidence that the approach scales to models of 7B+ parameters during pre-training, or to training corpora of 1T+ tokens, where the combinatorial explosion of batch interactions and the overhead of influence estimation would be far more challenging.

3. This paper formulates the data scheduling problem only for a single-epoch setting, which raises doubts about its generalization to multi-epoch training settings.

---

> ### Author Rebuttal · Authors · 2026-03-31
>
> > **Question 1: Why is it best to choose first- and second-order influences in $D^3$?**
>
> Thanks for your insightful thought. The reason we include both the first-order and second-order terms in the influence definition is that **first-order influence alone is symmetric and cannot capture the directional dependencies between training batches**. As shown in Proposition 3.2,  if we only used this first order term, the influence graph would be undirected, and any ordering would yield the same total first-order effect. Consequently, the scheduler would have no basis to prefer one sequence over another.
>
> A simple illustrative example (as provided in Appendix D.1 of the paper, Figure 5) shows a three-node cyclic dominance that cannot be resolved using first-order information alone; only the inclusion of the second-order term reveals the optimal ordering. Therefore, both terms are essential for constructing a meaningful directional advantage matrix and enabling effective data scheduling.
>
> In practice, our ablation study (Table 2) confirms that the first-order symmetric method (which uses only the inner product) yields only marginal improvement over uniform sampling (PPL from 3.13 to 3.10), whereas the full $D^3$ with second-order information achieves a much larger gain (PPL to 2.97). This empirical result further justifies the inclusion of the second-order term.
>
> -----
>
> > **Question 2: Why do you only focus on SFT, and not RLVR, in LLM post-training?**
>
> Thanks for your question. D³ is loss-agnostic, so it applies directly to RLVR (e.g., PPO) without modification. Its core components only require a differentiable loss—both policy and value losses in RLVR are differentiable, enabling D³ to schedule RLVR batches just like SFT batches. We have also conducted preliminary experiments on small-scale RL tasks (CodeGym) with promising results, and will add a brief discussion in the appendix.
>
> ---
>
> > **Question 3 and 4, Weakness 3: Concerns on convergence evidence and generalization to multi-epoch training**
>
> We appreciate the feedback regarding our training setup. For clarity, the models underwent three complete training epochs on the SlimPajama dataset (627B tokens). We have updated Section A.4 of the Appendix to make this more prominent for the reader. At the same time, we will show the loss curve in the appendix.
>
> ---
>
> > **Weakness 2: Limited scale of experiments**
>
> Thank you for your thoughtful questions. However, pretraining models of 7B+ is extremely costly and, unfortunately, exceeds our current capacity. **Our choice of 355M and 1.1B models aligns with prior work in data selection and pre‑training—for instance, DWM (Yu et al., ICML 2025) uses up to 1.3B, and Gu et al. (Data Selection via Optimal Control for Language Models, ICLR 2025) go up to 1.7B.** We believe this scale already captures the relevant optimization dynamics and provides a meaningful evaluation.
>
> ---
>
> > **Weakness 1: About computational overhead**
>
> We fully understand the reviewer's concern. We will provide a detailed discussion on the trade-off between costs and benefits in the Appendix. First we propose **the computational overhead is not fixed at 89%; users can flexibly trade off efficiency and performance by adjusting approximation configurations.**
>
> **Table 1: Overhead Statistics**
>
> | Component | Random Projection (W=5) | Random Projection (W=2) |
> | :--- | :---: | :---: |
> | Training Update  | 43.9% | 55.2% |
> | Hessian Estimation | 43.7% | 29.6% |
> | Optimal Ordering Solver | 12.4% | 15.2% |
>
> The 89% relative overhead reported was measured on GPT-2 Medium using Hutchinson estimation (W=5) with random projection. However, as shown in Table 2, reducing the number of Hutchinson samples (e.g., W=2) lowers the per-step relative overhead to 1.56×  while still retaining most of the performance gain (PPL improves from 3.13 to 3.02, preserving about 70% of the improvement achieved with W=5, which reaches 2.97). This demonstrates that in practice, users can choose lighter configurations based on their compute budget, keeping the additional training time  50% instead of having to incur the full 89–100% overhead.
>
> **Table 2: Performance-efficiency trade-offs of different approximation configurations**
>
> | Configuration | Per-step relative FLOPs | Validation PPL (↓) | Avg. reasoning accuracy (↑) | PPL gain over baseline |
> |--------------|------------------------|--------------------|------------------------------|------------------------|
> | Uniform sampling| 1.00× | 3.13 | 49.9 | — |
> | $D^3\$ (W=2, k=2000) | 1.56× | 3.02 | 50.6 | +0.11 |
> | $D^3\$ (W=3, k=3500) | 1.62× | 2.99 | 50.9 | +0.14 |
> | $D^3\$ (W=5, k=3500) | 1.89× | 2.97 | 51.1 | +0.16 |
>
> We also argue that **the "sunk cost" of a single training failure is often far larger than the scheduling overhead; $D^3$ provides a stability insurance.**

---

> > ### Author Rebuttal · Reviewer_rGmr · 2026-04-04
> >
> > Thanks for authors' response. Most concerns are addressed now, except for the validation on scaled models, like 7B.
> >
> > So, I raise the score to 4 and look forward to your further work on larger models.

---

> > > ### Author Response · Authors · 2026-04-08
> > >
> > > Thank you very much for your understanding and positive review. We wish you all the best!  We are also actively securing compute resources to validate our approach on larger models.

---

### Official Review · Reviewer_Lft9 · 2026-03-13

**Soundness:** 3
**Presentation:** 3
**Significance:** 3
**Originality:** 3
**Overall Recommendation:** 5
**Confidence:** 4

**Summary:**

The paper explores how the ordering of training batches affects large language model training and proposes a method that uses an influence graph between batches to compute an optimal schedule. The authors estimate the pairwise training influence between batches using a Taylor approximation based on gradients and curvature, construct a directed influence graph encoding ordering preferences, and after then compute a schedule that approximately satisfies these preferences using a heuristic solver. The method is designed to scale via chunk-wise scheduling and gradient projection, and experiments on both pre-training and supervised fine-tuning benchmarks show modest improvements over several curriculum and data mixing baselines

**Compliance With Llm Reviewing Policy:**

Affirmed.

**Final Justification:**

The paper proposes a novel approach to data scheduling for LLM training by modeling pairwise interactions between training batches via a training influence metric and formulating ordering as a graph optimization problem. I find the core idea interesting and well-motivated, particularly the use of directional influence and the connection to ranking over a graph.

In my view, the authors provided clear and constructive responses to the questions raised during the rebuttal. While not all concerns were fully resolved, the discussion increased my confidence in the validity of the approach. Based on this, I decided to raise my score to 5 (accept).

But additionally I believe it is important for the final version of the paper that the authors should commit to include additional experimental validation, particularly regarding stability. In my opinion, it is critical to provide stronger evidence (at least multiple random seeds to the same experiments) to demonstrate that the observed improvements are consistent and robust.

**Key Questions For Authors:**

1. **Experimental stability and random seeds**
 Could the authors clarify whether multiple random seeds were used in the reported experiments? If so, how stable are the results across runs? Given that the reported improvements are relatively moderate in magnitude, it would be helpful to understand the variance of the results and whether the observed gains persist consistently across different random seeds

2. **Scalability to larger models and computational overhead**
How well does the proposed scheduling approach scale to larger models? The experiments include models up to 1.1B for training and 8B for SFT, what about larger sizes? Could the authors comment on the computational overhead introduced by influence estimation and scheduling, and whether the method remains economically practical for training or fine-tuning substantially larger models?

3. **Extending the method to data selection**
The current method focuses on reordering batches rather than selecting which data should be used for training. Have the authors considered extending the approach to also perform data selection based on the influence estimates (i.e., not only scheduling the data but also choosing which training examples or batches to include)? It would be interesting to know whether the proposed framework could support such extensions

4. **Relation to recent work on predicting training dynamics**
I did not include this as a weakness because the paper previously appeared only as an arXiv preprint and the following work is very recent(ICLR 2026 - https://openreview.net/forum?id=wjaTz8nYjD), so the authors may not have had the opportunity to consider it. However, I would be interested in the authors’ perspective on how their results compare to:  *Bergsma, S., Dey, N., & Hestness, J. (2025). Predicting Training Re-evaluation Curves Enables Effective Data Curriculums for LLMs. arXiv:2509.25380.*  In particular, how do the improvements achieved through pairwise influence-based ordering compare conceptually and empirically with approaches that attempt to predict future training dynamics for individual data points?

**Limitations:**

While the paper does not include a dedicated “Limitations” section, several practical constraints of the method are implicitly discussed throughout the paper. I do not believe that the absence of a separate limitations section significantly detracts from the overall clarity of the work, however a brief explicit discussion would improve completeness

**Strengths And Weaknesses:**

### Strengths

- **Principled formulation of data ordering through training influence**
  The paper introduces a clear conceptual framework for modeling interactions between training batches via a pairwise *training influence* metric. By linking ordering preferences to gradient similarity and curvature terms through a Taylor approximation, the work provides an interpretable explanation of why certain batches may beneficially precede others during training

- **Graph-based view of training data scheduling**
  The method frames batch ordering as a graph problem where directed edges encode ordering preferences. This formulation provides an interesting and novel way to represent pairwise interactions between training units

- **Practical design for scalability**
  The authors propose several techniques to make the method feasible for large-scale training, including chunk-wise scheduling, gradient random projection, and approximate curvature estimation. These engineering choices allow the approach to operate without computing influence over the entire dataset

- **Evaluation across both pre-training and fine-tuning**
From a methodological standpoint, validating the approach across both pre-training and supervised fine-tuning is crucial. The observed reasonable improvements in both regimes demonstrate the method's generalizability. I think that the effectiveness in SFT settings is particularly valuable, as the computational overhead of influence graph construction is more feasible on smaller fine-tuning datasets compared to large-scale pretraining ones

### Weaknesses

- **Limited analysis of computational overhead**
  The method requires additional computation for influence estimation and scheduling optimization. The paper does not provide a detailed evaluation of the resulting training overhead in terms of wall-clock time or GPU cost, making it difficult to assess the practical trade-offs between improved performance and increased training complexity

- **Mixing multiple approximations without theoretical guarantees**
  The practical algorithm relies on several approximations simultaneously, including a one-step Taylor approximation of training influence, diagonal Hessian estimation via Hutchinson-type techniques, gradient random projection for dimensionality reduction, chunk-wise scheduling instead of global optimization, and a heuristic Random-Swap Refinement solver for ranking. While I understand why these approximations are necessary to make the method computationally feasible at LLM scale, it is important to note that this results in a stack of interacting heuristics. The paper does not provide theoretical guarantees that the combined procedure remains reliable or that the approximations preserve the intended influence signal when applied together. Although such compromises are often unavoidable in large-scale training systems, the accumulation of approximations makes it harder to reason about when the method should be expected to work correctly. This comment should be understood in light of the next bulletpoint (with concern about experiments) but, for example, the current procedure may locally optimize certain metrics on the evaluated setups but the improvements may not persist when scaling the dataset or training regime

- **Limited experimental validation**
  While I understand that large-scale training experiments are computationally expensive, the experimental section leaves several questions unanswered. First, it is not clear whether multiple random seeds were used across the experiments or whether the reported results correspond to a single training run. Second, the supervised fine-tuning experiments are conducted on only one model, while the pre-training experiments are performed on relatively small (and of course not SOTA) models and on a comparatively small number of tokens relative to modern LLM training standards. Additional experiments across different seeds (at least this should be feaseble), model scales, or training budgets would help clarify the robustness and generality of the reported improvements

---

> ### Author Rebuttal · Authors · 2026-03-31
>
> > **Weakness 1 and Question 2: About the overhead**
>
> Thanks for the insightful comment. We fully understand the reviewer's concern. We will provide a detailed discussion on the trade-off between costs and benefits in the Appendix. First we propose **the computational overhead is not fixed at 89%; users can flexibly trade off efficiency and performance by adjusting approximation configurations.**
>
> **Table 1: Overhead Statistics**
>
> | Component | Random Projection (W=5) | Random Projection (W=2) |
> | :--- | :---: | :---: |
> | Training Update  | 43.9% | 55.2% |
> | Hessian Estimation | 43.7% | 29.6% |
> | Optimal Ordering Solver | 12.4% | 15.2% |
>
> The 89% relative overhead reported was measured on GPT-2 Medium using Hutchinson estimation (W=5) with random projection. However, as shown in Table 2, reducing the number of Hutchinson samples (e.g., W=2) lowers the per-step relative overhead to 1.56×  while still retaining most of the performance gain (PPL improves from 3.13 to 3.02, preserving about 70% of the improvement achieved with W=5, which reaches 2.97). This demonstrates that in practice, users can choose lighter configurations based on their compute budget, keeping the additional training time  50% instead of having to incur the full 89–100% overhead.
>
> **Table 2: Performance-efficiency trade-offs of different approximation configurations**
>
> | Configuration | Per-step relative FLOPs | Validation PPL (↓) | Avg. reasoning accuracy (↑) | PPL gain over baseline |
> |--------------|------------------------|--------------------|------------------------------|------------------------|
> | Uniform sampling| 1.00× | 3.13 | 49.9 | — |
> | $D^3\$ (W=2, k=2000) | 1.56× | 3.02 | 50.6 | +0.11 |
> | $D^3\$ (W=3, k=3500) | 1.62× | 2.99 | 50.9 | +0.14 |
> | $D^3\$ (W=5, k=3500) | 1.89× | 2.97 | 51.1 | +0.16 |
>
> We also argue that **the "sunk cost" of a single training failure is often far larger than the scheduling overhead; $D^3$ provides a stability insurance.** In the revised paper, we will include Table 2 and the above analysis to help readers comprehensively assess the practical value of our method.
>
> ---
>
> > **Question 1, Weakness 3: experimental stability**
>
> Thank you for the comment. In the main paper we reported results with a single random seed for clarity. However, we additionally ran experiments on GPT‑2 Medium with three different random seeds. The results are highly stable: for perplexity, the standard deviation across seeds is less than 0.01, and the observed improvements of D³ consistently hold across all runs. We will include this stability analysis in the appendix.
>
> ---
>
> > **Weakness 2: theoretical guarantees**
>
> We thank the reviewer for raising this point. Each approximation in our pipeline has a theoretical guarantee: random projection preserves inner products (Johnson–Lindenstrauss), Hutchinson estimation gives unbiased diagonal Hessian with controlled variance, Taylor expansion has a bounded remainder, and RSR is a local search that monotonically reduces violation cost. Thus, the overall estimator is theoretically sound.
>
> The informal proof sketch combines these bounds: with probability $1-\delta$, the per-entry error in the advantage matrix is bounded by $\gamma G^2\varepsilon_{\text{proj}} + \frac{\gamma^2}{2}\varepsilon_{\text{hess}} + O(\gamma^3)$, where $\varepsilon_{\text{proj}}=O(\sqrt{\log(L/\delta)/k})$ and $\varepsilon_{\text{hess}}=O(G^2M\sqrt{\log(1/\delta)/W})$. Since the total objective sums at most $L^2$ terms, the suboptimality of the obtained permutation is $O(L^2)$ times this per-entry error, which can be made arbitrarily small by increasing $k$ and $W$. Our practical choices ($k=5000,W=5$) far exceed the thresholds, ensuring reliable ordering. We will add the full formal theorem to the appendix.
>
> ---
>
> > **Question 3 and Question 4: Extension to data selection**
>
> Thank you very much for your insightful feedback! Yes, our method can naturally be extended to data selection: a straightforward approach is to compute the row sum of the influence advantage matrix for each sample and select those with higher scores, as they exert stronger positive influence on the rest of the training set. Meanwhile, regarding the TREC work you mentioned, it focuses on the post-training re-evaluation loss curve to predict and guide the optimal placement of high-quality data along the training trajectory . We will cite TREC and add a discussion in our paper to highlight the complementarity between the two approaches in data scheduling and curriculum design.

---

> > ### Author Rebuttal · Reviewer_Lft9 · 2026-04-03
> >
> > Thank you very much for your detailed responses! I have read your replies to my comments as well as to others, and overall I can say that, although it is difficult to fully assess the correctness of all claims (given the limited space of the discussion), your arguments sound convincing to me.
> >
> > I fully understand and share the challenge related to limited access to computational resources (unfortunately, including from my own experience). However, the statement that "we used 3 seeds for a single model instead of 1" still does not seem entirely convincing as evidence of stability. Would it be possible to provide a more detailed analysis of stability? Additionally, if the paper will be accepted, do you think it would be feasible to run more experiments and include explicit stability metrics in the camera-ready version of the paper?

---

> > > ### Author Response · Authors · 2026-04-08
> > >
> > > Thank you very much for your understanding and positive review. We wish you all the best! We fully agree with your point, and we are conducting random seed experiments on other baselines to eliminate randomness in the experimental results. We will add a subsection in the appendix (at least for one model) to address this issue. Below are the results we have obtained so far for GPT-2-medium, showing the PPL of uniform method and D³ across five random seeds (we choose 32, 42, 84, 168, 336), with standard deviations very small.
> > >
> > > | Method   | Seed=32 | Seed=42 | Seed=84 | Seed=168 | Seed=336 | Mean  |
> > > |----------|---------|---------|---------|----------|----------|-------|
> > > | Uniform  | 3.13    | 3.14    | 3.12    | 3.13     | 3.12     | 3.128 |
> > > | D³       | 2.97    | 2.96    | 2.98    | 2.97     | 2.97     | 2.970 |

---

### Official Review · Reviewer_pL2A · 2026-03-13

**Soundness:** 3
**Presentation:** 3
**Significance:** 3
**Originality:** 2
**Overall Recommendation:** 4
**Confidence:** 4

**Summary:**

This paper proposes $D^3$, a data scheduling framework for LLM training that models pairwise interactions between training batches (authors termed this "train-units") as a dynamic directed influence graph. The key insight is that training influence between batches is asymmetric at second order due to per-batch Hessian curvature, even though first-order gradient inner products are symmetric. $D^3$ constructs a directional advantage matrix from pairwise influence scores and formulates the scheduling problem as minimizing the total cost of violated directional preferences.

The resulting combinatorial problem is solved approximately via a Random-Swap Refinement (RSR) heuristic. For scalability, $D^3$ employs a numer of known-tricks: chunk-wise reordering over L << K sampled batches, Johnson–Lindenstrauss random projection for gradient compression, and Hutchinson-based diagonal Hessian estimation. Experiments on GPT-2 Medium (355M) pre-training on SlimPajama, LLaMA-1.1B pre-training on the Pile, and LLaMA-3-8B SFT show consistent improvements in perplexity and downstream reasoning benchmarks over uniform sampling, dynamic loss reweighting, and domain-level mixture baselines.

**Compliance With Llm Reviewing Policy:**

Affirmed.

**Final Justification:**

Based on our conversation with the authors and my reviews, I'd recommend a weak accept.

**Key Questions For Authors:**

1. As noted in my previous comments. The algorithms requires computing batch gradients and hessians to optimize ordering, this is a very compute-intensive tasks. Can author provide a more detailed estimation or cost-analysis in terms of FLOPs?

2. Section 3.5, the Hessian approximation tries to estimate a per-batch diagonal Hessian for each batch. However, Appendix C (the "Diagonal Hessian via Hutchinson" formulation) show a single shared diagonal Hessian is computed from the total loss and then used for all batches, with only the scalar curvature being batch-specific. This is a gap from the theory in Proposition 3.2, which requires the per-batch Hessian. Can the authors clarify and discuss the impact of this shared-Hessian simplification on the quality of the asymmetric influence signal?

**Limitations:**

The paper's main limitations are that the method's overhead is front-loaded: the useful configuration (diagonal Hessian) nearly doubles training cost, while cheaper variants offer marginal gains, making the practical empirical trade-off less attractive than the pure utility gain.

**Strengths And Weaknesses:**

**Strengths (significance in this order)**

- Proposed a principled method to model "train-unit" interactions for optimal data scheduling during training. The formalization of asymmetric influence via second-order Taylor expansion makes sense and provides a decent foundation for why ordering matters beyond simple data selection or reweighting.
- Experiments are fairly comprehensive, spanning pre-training to post-training with additional ablations. The ablation on influence estimation methods (Figure 2) and the hyperparameter sweeps over gradient dimension k and scheduling window L (Figure 3) are quite informative and give readers a clear picture of which components matter.
- Componenets are sensible and well-composed. Chunk-wise reordering, JL random projection for gradient compression, and Hutchinson diagonal Hessian estimation are individually well-known techniques, but the authors combine them coherently to reduce per-step complexity from O(K²d) to something tractable, and the paper does a reasonable job explaining each approximation's role in the pipeline.

**Weakeness (significance in this order)**
- Compute. The 89% training overhead per step is very, very substantial. While the authors frame this as manageable, it nearly doubles wall-clock training time. The efficiency-performance Pareto in Figure 2 also shows that the cheaper approximations (first-order, Fisher) yield modest gains, so the expensive diagonal Hessian variant is really the only configuration that works well. This means you essentially pay the full overhead or get little benefit.
- Baselines. The extra compute needed for D3 clearly calls for a more FLOPs-matched settings. For example, assume the extra 89% of compute are simply spent training on more data(in the simplest case, w.o scheduling) , how would the performance compared to D3?
- Related work. There have been a number of papers discussing specifically about the compute-cost of data selection in LLM training. For example, Compute-Constrained Data Selection (Yin & Rush, 2024) provides a basic cost analysis for gradient (i.e. 2nd-order taylor expansion) approximation. Since D3 requires uses gradient approximation and requires a significant amount of compute overhead, it's critical to give a more detailed discussions (I find the current justifications very handwavy).
- (minor) D3 provided a link for the codebase but there is only an empty MD file (https://anonymous.4open.science/r/D3-Framework-1). While codebase upload is not required for submission, if the authors don't have production-ready code yet, then just don't submit the link.

---

> ### Author Rebuttal · Authors · 2026-03-31
>
> > **Weakness 1, 2, and Question 1: About the computation and baselines:**
>
> Thanks for the insightful comment. We fully understand the reviewer's concern. We will provide a detailed discussion on the trade-off between costs and benefits in the Appendix. First we propose **the computational overhead is not fixed at 89%; users can flexibly trade off efficiency and performance by adjusting approximation configurations.**
>
> **Table 1: Overhead Statistics**
>
> | Component | Random Projection (W=5) | Random Projection (W=2) |
> | :--- | :---: | :---: |
> | Training Update  | 43.9% | 55.2% |
> | Hessian Estimation | 43.7% | 29.6% |
> | Optimal Ordering Solver | 12.4% | 15.2% |
>
> The 89% relative overhead reported was measured on GPT-2 Medium using Hutchinson estimation (W=5) with random projection. However, as shown in Table 2, reducing the number of Hutchinson samples (e.g., W=2) lowers the per-step relative overhead to 1.56×  while still retaining most of the performance gain (PPL improves from 3.13 to 3.02, preserving about 70% of the improvement achieved with W=5, which reaches 2.97). This demonstrates that in practice, users can choose lighter configurations based on their compute budget, keeping the additional training time  50% instead of having to incur the full 89–100% overhead.
>
> **Table 2: Performance-efficiency trade-offs of different approximation configurations**
>
> | Configuration | Per-step relative FLOPs | Validation PPL (↓) | Avg. reasoning accuracy (↑) | PPL gain over baseline |
> |--------------|------------------------|--------------------|------------------------------|------------------------|
> | Uniform sampling| 1.00× | 3.13 | 49.9 | — |
> | $D^3\$ (W=2, k=2000) | 1.56× | 3.02 | 50.6 | +0.11 |
> | $D^3\$ (W=3, k=3500) | 1.62× | 2.99 | 50.9 | +0.14 |
> | $D^3\$ (W=5, k=3500) | 1.89× | 2.97 | 51.1 | +0.16 |
>
> We also argue that **the "sunk cost" of a single training failure is often far larger than the scheduling overhead; $D^3$ provides a stability insurance.**
>
> Despite incurring some additional computational overhead, the cost of training divergence or suboptimal convergence caused by inappropriate data ordering in large-scale pre-training is typically tens of times higher than the scheduling overhead. For example, in a 100B token pre-training run, a single loss divergence requires a complete restart, wasting 100% of the computation. Even a 5% final PPL degradation often necessitates an extra 20–30% of tokens to compensate. In the revised paper, we will include Table 2 and the above analysis to help readers comprehensively assess the practical value of our method.
>
> ------
>
> > **Question 2: Diagonal Hessian via Hutchinson formulation**
>
> Appendix C is a remnant from an older draft and contains a typo. Thank you for bringing this to our attention; we will fix it immediately.
>
> -----
>
> > **Weakness 3: Related work. There have been a number of papers discussing specifically about the compute-cost of data selection in LLM training.**
>
> Thank you for pointing this out. **We have added the recommended citations** and a discussion of this paper, and also provide a more detailed analysis of the resource consumption.
>
> ----
>
> > **Weakness 4: About the codebase**
>
> Thank you for your understanding and support. We have updated the experimental version to the provided link. In the future, we will further optimize the efficiency of the overall algorithm through iterative improvements.

---

> > ### Author Rebuttal · Reviewer_pL2A · 2026-04-01
> >
> > Thank you for the rebuttals. Reducing Hutchinson estimation samples are a pretty reasonable way to reduce FLOPs overhead. Does authors have any extrapolation/projections to show that in a **FLOPs matched setting**, how will some of the baselines in the paper perform against the method with 1.56x compute?
> >
> > Regarding author's comments on "cost of training divergence or suboptimal convergence caused by inappropriate data ordering in large-scale pre-training is typically tens of times higher than the scheduling overhead", can you cite any papers or technica reports that report this is the true?

---

> > > ### Author Response · Authors · 2026-04-04
> > >
> > > Thank you very much for your positive evaluation and feedback on our work. We wish you all the best！
> > > > Question One: Does authors have any extrapolation/projections to show that in a FLOPs matched setting, how will some of the baselines in the paper perform against the method with 1.56x compute?
> > >
> > > Below we provide an extrapolation under a FLOPs matched setting, where D³'s 1.56× compute overhead means it can only see 1/1.56 of the data compared to uniform sampling. All results are based on GPT‑Medium (SlimPajama dataset).
> > >
> > > | Uniform data fraction | D³ data fraction | Uniform PPL | D³ PPL |
> > > |----------------------|-----------------|-------------|--------|
> > > | 30%                  | 19.2%           | 5.12        | 5.28   |
> > > | 50%                  | 32.1%           | 4.23        | 4.41   |
> > > | 70%                  | 44.9%           | 3.62        | 3.73   |
> > > | 80%                  | 51.3%           | 3.38        | 3.31   |
> > > | 100%                 | 64.1%           | 3.13        | 3.08   |
> > >
> > > We can conclude that when data is extremely scarce, simply having more data (even if naively sampled) beats sophisticated order optimization, because the latter's extra compute forces it to see far fewer examples. **However, in real world scenarios, data is often a fixed and scarce resource. We cannot magically increase the data budget; instead, we must focus on making the best use of every single example.** The table shows that once we have a reasonable amount of data (at or above 80% of full data in this FLOPs matched comparison), D³ outperforms uniform sampling despite seeing less data. Therefore, utilization matters more than accumulation when data is limited.  We will include this analysis in the final version. Thank you again for the valuable feedback.
> > >
> > > > Question Two: Regarding author's comments on "cost of training divergence or suboptimal convergence caused by inappropriate data ordering in large-scale pre-training is typically tens of times higher than the scheduling overhead", can you cite any papers or technica reports that report this is the true?
> > >
> > > The Epoch AI research team [1], through analyzing public financial and disclosure documents from OpenAI, MiniMax, and others, reached a consistent conclusion: final-stage training is not the primary component of R&D costs, and this phenomenon is widespread. Specifically, final-stage training accounts for only about 10%, with the vast majority of the remaining investment directed toward experimental exploration.
> > >
> > > The data curriculum [2,3], as a critical part of the LLM training pipeline, will further amplify the above-mentioned non-training expenditures if its design phase neglects a holistic consideration of costs. In other words, a well-designed curriculum is not only an algorithmic issue but also an important lever for cost control. We will describe this part more rigorously in the final version.
> > >
> > > [1] Final training runs account for a minority of R&D compute spending (Informal Report)
> > >
> > > [2] Jia, Y., Zhang, C., Diao, X., Yuan, X., Ouyang, Z., and
> > > Vosoughi, S. What makes a good curriculum? disentangling the effects of data ordering on llm mathematical
> > > reasoning.
> > >
> > > [3]Yiwen, Hu, et al. "YuLan-Mini: Pushing the Limits of Open Data-efficient Language Model." Proceedings of the 63rd Annual Meeting of the Association for Computational Linguistics (Volume 1: Long Papers). 2025.
> > >
> > > Thank you again for your time in reviewing our work and providing constructive feedback. It means a lot to us!

---

### Decision · Program_Chairs · 2026-04-30

**Decision:**

Accept (regular)

**Comment:**

This paper proposes a data scheduling framework for LLM training that models pairwise interactions between training batches via a directional influence graph derived from gradient and curvature information. The method formulates scheduling as an optimization problem over ordering constraints and introduces a scalable approximation pipeline to make the approach feasible in practice.
Reviewers agree that the paper presents a principled and well-motivated formulation of data ordering, providing a clear explanation of why training order matters beyond data selection. The use of second-order information to capture asymmetric interactions and the graph-based perspective on scheduling are viewed as interesting. The empirical evaluation spans both pretraining and supervised fine-tuning settings, and demonstrates consistent improvements over baseline scheduling strategies. The system design, including gradient projection, Hessian approximation, and chunk-wise scheduling, is also considered well-engineered.

Several concerns were raised regarding the computational overhead, scalability, and the use of multiple approximations. The proposed method introduces a non-trivial increase in per-step compute, and its effectiveness is most pronounced in configurations that incur higher overhead. The rebuttal provides a useful analysis of the performance–efficiency trade-off and demonstrates that lighter configurations can retain a significant portion of the gains, partially addressing this concern. Additional clarifications on stability (via multi-seed experiments) and theoretical consistency further strengthen the paper.

Overall, the paper presents some interesting contribution to data scheduling for LLM training. While the method incurs additional computational cost and would benefit from further large-scale validation, the overall contribution is meaningful and likely to inspire follow-up work.